# SaTran: An efficient Transformer exploiting Spatiotemporal Redundancies for Satellite Image Time Series Representation Learning

## Abstract

Earth observation applications like crop yield prediction, solar energy prediction, land cover classification, etc., need large size Satellite Image Time Series (SITS) leading to huge computational requirements. A couple of BERT-based models exist which work at pixel level unable to exploit spatial correlation among pixels and also require ground truth at pixel granularity during fine-tuning, rendering them infeasible for prediction tasks. The models based on Vision Transformer factorize spatial and time dimensions and first process images and then time series of image embeddings. However, in many cases, SITS require simultaneous analysis of both dimensions. We present a transformer, SaTran, which focuses on non-redundant patch tubes to overcome the limitations listed above. Transformers developed for RGB videos are found lacking when applied to SITS data characterized by the presence of patches with spatiotemporal redundancy persisting throughout the time series. SITS data also has patches where temporal redundancy lasts only for a few timestamps. The salient features of SaTran include: 1) an automatic patch tube selection mechanism which ignores spatiotemporally redundant patches; 2) exploitation of spatial correlation between pixels by the processing of patch tubes and handling of their temporal redundancy using tube masking; 3) two-fold handling of redundancy and distributed application of VideoMAE enables space and time efficient processing of large size SITS; and 4) learning end task agnostic representation of entire time series. Extensive experimentation shows that SaTran outperforms competing models and exhibit state-of-the-art performance for various earth observation applications. The code is available on (.. will be given after acceptance..).

## 1 Introduction

Satellite Image Time Series (SITS) data offer valuable insights into Earth's surface characteristics and dynamics. It has widespread applications across domains like ecology, agriculture, forestry, land management, disaster monitoring, risk assessment, etc. Deep learning has gained popularity in the remote sensing community due to its ability to learn valuable features from input data without feature engineering. In the realm of SITS classification, a combination of convolutional and recurrent neural networks are employed to capture spatiotemporal characteristics from the data. An alternative to RNNs, transformers, originally proposed for natural language processing tasks, have shown promising performance in sequence encoding. A couple of studies Yuan & Lin (2020); Yuan et al. (2022) used BERT to classify time series for every pixel which limits them from effectively exploiting the spatial correlations in the image time series. Moreover, these models work only for classification as they segment the image time series which is not suitable for prediction problems like prediction of crop yield, snow cover, cloud cover, etc. The significance of these applications is given in Appendix B.1. In prediction problems, the ground truth is mostly available for coarser granularity than that of pixel e.g., at a county or a district level. Another model TSViT Tarasiou et al. (2023) used ViT for landcover classification. The authors factorize input dimensions into spatial and temporal components to reduce the computation. However, the model is not able to identify the redundancy in the patches and processes them all.

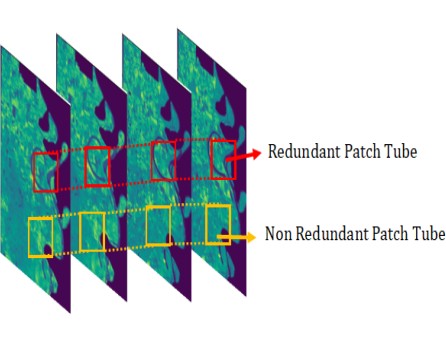

Figure 1: Types of Patch Tubes

Satellite data is much larger than the large datasets used in NLP or vision domains Rolf et al. (2024). The size of an image for Landsat-8 satellite, a high spatial resolution satellite, is approximately $2000 \times 2500 \times \#bands$. For yearly time series of Landsat-8, the size becomes $2000 \times 2500 \times 5(\#bands) \times 23(\#timestamps)$ which makes its data volume $\approx$ 700MB. The total data used in the study for pre-training and downstream tasks comprises of around 2000 counties making the amount of data is 10.0 TB for Landsat-8 which is much larger than the datasets used in other domains Nast. Processing single image time series of Landsat-8 using existing video vision models leads to GPU out-of-memory (OOM) error on A100 GPU card with 80GB memory.

There are a few models like AdaVit Meng et al. (2022), and DynamicVit Rao et al. (2021) which reduce the computational requirements by processing images in patches and using attention to select only informative patches. Authors Hou et al. (2022) used the token dropping mechanism to ignore less informative tokens while pre-training BERT. However, as mentioned Esther Rolf et. al. Rolf et al. (2024) any models developed for 3-channel RGB images or videos are suboptimal for modeling satellite datasets due to the large volume and different characteristics of SITS data. Moreover, applications like crop yield prediction, soil moisture prediction, etc. require time series of large size images to be processed leading to computational bottleneck. Thus, we require a method which can train a Large SITS Model efficiently.

We propose a transformer model, SaTran, for large size satellite image time series which exploits spatiotemporal redundancies. SITS data can be characterized by the presence of patches with spatiotemporal redundancy persisting throughout the time series, referred to hereafter as redundant patch tubes. SITS data also contains patches where temporal redundancy lasts only for a few timestamps, referred to hereafter as non-redundant patch tubes. The pictorial representation of the classification of patch tubes is given in Figure 1. For example, a region of a barren land/water body has spatiotemporal redundancy, and it won't change even for years (thus is a redundant patch tube); 2) the non-redundant patches (regions of interest) experience changes with time but can still have a temporal redundancy for a shorter span, for example, cultivation land experiences changes in the crop cycle duration. However, during harvest time when the crop is fully grown, there can be redundancy for a few time stamps. Removing redundancies reduces the computational requirements thereby helping in the democratization of satellite image technology. SaTran disentangles spatiotemporal and temporal redundancies and makes the SITS processing efficient. Its key features are:

1. Dual Redundancy-Handling Mechanism: SaTran introduces two novel modules to address specific challenges in SITS which were not there in the baseline transformer:

   - PatchTubeSelect: Selects non-redundant patch tubes by leveraging an attention mechanism, which reduces input size by focusing only on spatiotemporal hotspots.
   - TemporalRedundancyHandler: Employs VideoMAE Tong et al. (2022) in distributed manner to effectively capture and process temporal patterns within these reduced hotspots, minimizing temporal redundancies and improves efficiency.

   Unlike standard Transformers, which operate on complete data and face scalability issues, SaTran's architecture reduces the computational burden by intelligently selecting and processing a subset of the data in distributed manner i.e., all the patch tubes are processed in parallel without compromising task performance and improves efficiency.

2. Task-Specific Adaptability: SaTran is designed with inherent flexibility to accommodate the diverse resolutions, temporal frequencies, and spectral characteristics of satellite data from systems like MODIS, Landsat-8, and Sentinel-2. The ability to tune hyperparameters such as patch size, traversal ratio, and attention thresholds allows SaTran to adapt to a wide variety of SITS tasks, such as land cover classification, change detection, etc.

3. Computational Efficiency: By selecting only top k critical patch tubes, SaTran processes significantly fewer tokens than standard Transformers, leading to reduced memory usage and runtime. This efficiency is critical for large-scale SITS datasets. Moreover, these patch tubes are independent and can be processed in parallel.

4. Scalability: SaTran's modular design scales well with higher-resolution data or longer time spans by keeping the number of selected patches manageable. It ensures practical feasibility for real-world applications, where datasets can grow exponentially in size.

5. We have done extensive experimentation and compared our model with existing video vision models and SITS models. The results show that SaTran outperforms the existing models for various downstream earth observation applications like crop yield prediction, snow cover prediction, land cover classification, etc.

SaTran is a generic and adaptive transformer which can be used for analytics of SITS obtained from any satellite system by adjusting the hyper-parameters like patch size, no. of channels, traversal ratio, etc. While going from low resolution to high resolution, the no. of patch tubes increases for a scene, however our approach requires only a partial no. of patch tubes to be processed. And thus, the computational requirement increases marginally. The perquisite of the proposed transformer is that it successfully and efficiently processes the large-size SITS obtained from high-resolution satellites (like Landsat-8) where existing vision models shortfall.

## 2 RELATED WORK

**Satellite Data:** Advancements in satellite image technology are leading to notable enhancements in capturing data with high spatial, temporal, and spectral resolutions. This progress enables effective monitoring of the Earth's surface at desired levels of detail, catering to various Earth observation applications. Popular satellite systems are MODIS NASA (2015), Landsat-8/9 NASA (2016), and Sentinel-2 European Space Agency Signature (2017) due to their publicly available data which can be used in different real-world applications like disaster management, urban planning, agriculture, climate studies, etc. This makes satellite imagery a cost-effective solution for obtaining global-scale data. Time series analytics are commonly required for many applications leveraging satellite imagery. Satellite image time series is analogous to RGB videos but it has characteristics that differentiate it from RGB videos e.g. SITS do not contain moving objects, the landscape does not change their position and only the changes are observed in them with time. This is the reason why the existing video models are not suitable for SITS data.

**Earth Observation Applications:** Researchers have tried to model SITS data in different forms according to specific applications. They have used satellite data in the form of histograms Sun et al.; You et al. (2017); Sun et al. (2020); Kaur et al. (2022), spectral reflectance indices Schwalbert et al. (2020); Hunt et al. (2019) or images Qiu et al. (2020) for different earth observation applications. Different studies have applied various statistical Tefera et al. (2022) and machine learning Johnson et al. (2016); Kern et al. (2018); Nagy et al. (2021); Vannoppen & Gobin (2021); Shammi & Meng (2021) on different types of data for crop yield prediction. Similarly, authors used satellite data for snow cover prediction Kaur et al. (2023) and wildfire prediction Gupta et al. (2023), respectively.

**Transformers in other domains:** Transformers are gaining success in areas of NLP and vision. BERT Devlin et al. (2018) is a revolutionary text model that has significantly advanced field of language understanding. Similarly, Video Vision Transformer (ViViT) Arnab et al. (2021) represents a significant advancement in domain of computer vision, extending the transformer architecture to handle sequential frames of video data. VideoMAE Tong et al. (2022) customized video tube masking approach characterized by an exceptionally high masking ratio to extract effective video representations during pre-training process. Authors Wang et al. (2023) introduced making the decoder of VideoMAE and proposed model VideoMAE:v2 to improve the accuracy of video analytics. Lingchen et al. Meng et al. (2022) proposed an Adaptive Vision Transformer (AdaViT) which pro-

cesses an image in patches. It learns which patches to use and which self-attention heads to activate for every image and thus reduces the computation cost. Similarly, Rao et al. proposed DynamicViT Rao et al. (2021) which prunes redundant tokens dynamically based on the input. The model divides the RGB images into independent patches and uses the attention masking concept to mask the tokens of patches which are of minimum importance. In another study Hou et al. (2022), Hou et al. proposed the token-dropping concept for accelerating the pertaining process of BERT. These models can be explored for their use in SITS analytics but are not directly applicable to satellite data.

**Models in SITS Analyisis:** As stated in Rolf et al. (2024), the basic properties of two types of data-regular RGB videos and SITS are different. Unlike RGB videos, the objects in temporal sequences of satellite images remain in a fixed position but change in appearance over time. Researchers Yuan & Lin (2020); Yuan et al. (2022) tried to adapt BERT for SITS classification at pixel level. Authors Yuan & Lin (2020) presented a self-supervised pre-training approach of BERT designed to initialize a transformer-based network. The model is tasked with predicting randomly contaminated observations within an entire time series of a pixel. Similarly, authors Yuan et al. (2022) extended the above work to apply BERT to the time series of the immediate neighboring pixels and then predict the label of the center pixel using SITSFormer. The use of BERT on pixel time series shown to be a potential method for improving SITS classification performance and mitigating overfitting challenges in the application. Tarasiou et al. Tarasiou et al. (2023) proposed Temporo-Spatial Vision Transformer (TSViT) which is based on famous ViT model and adopted for satellite image time series analytics. TSViT divides a SITS into non-overlapping patches across both spatial and temporal dimensions which are then tokenized and processed by a factorized temporal-spatial encoder. However, these models are not suitable for applications like prediction of crop yield, snow cover, cloud cover, etc. where the ground truths cannot be available at pixel level. For such applications, the ground truths are available at a bigger region like a county or district and it necessitates processing the data at the image time series level and not at pixel time series level.

The existing models for SITS are designed specifically for classification problems and are not able to solve prediction tasks. The video analytics models are not suitable to be adapted for SITS. The proposed model in this paper works to resolve these two problems.

# 3 PROPOSED MODEL: SATRAN

In this section, we discuss the characteristics of satellite image time series and how it is different from RGB videos, and then describe the architecture of SaTran to handle challenges posed by SITS.

## 3.1 CHARACTERISTICS OF SITS

Satellite image data are stored in a raster format, organized as a tensor with dimensions for height, width, and channels. Temporal aspects can be integrated by arranging spatially-aligned rasters along a fourth dimension. Although this structure resembles images\videos, satellite images are far different from their equivalent in natural images for many reasons including:

1) Number of channels in satellite images and size of images from high spatial resolution satellites.

2) Unlike RGB videos, SITS often exhibit spatiotemporal redundancy over time, especially in specific landforms. For example, water bodies will have consistent patterns year after year. The spatial arrangements of landforms in SITS data do not change drastically over time. In contrast, RGB videos capture dynamic scenes where spatial configurations change frequently. The stable patterns observed in SITS contrasts with the fluid nature of video imagery, especially in urban environments or areas with constant human activity leading to huge spatiotemporal redundancy.

3) The rate of temporal redundancy is different in different spatial landforms. The redundant patch tubes such as water bodies, urban area, etc. will have temporal redundancy for a longer time. However, non-redundant patch tubes like areas under snow experience different redundancy for a shorter span. In regions where snowfall is a regular occurrence, such as high-altitude areas or northern latitudes, temporal redundancy in satellite image time series can be less during initial phase which can eventually be relatively high during winter months once snow settles and decreases as temperature rises. This leads to dynamic changes in the landscape. The once uniform snow cover gives way to patches of melting snow, revealing underlying terrain and vegetation. During this transition period, the temporal redundancy in satellite imagery decreases as the spatial patterns evolve rapidly.

4) SITS do not contain moving objects and thus the orientation of regular RGB videos is of more importance and SITS don't have any natural orientation Rolf et al. (2024).

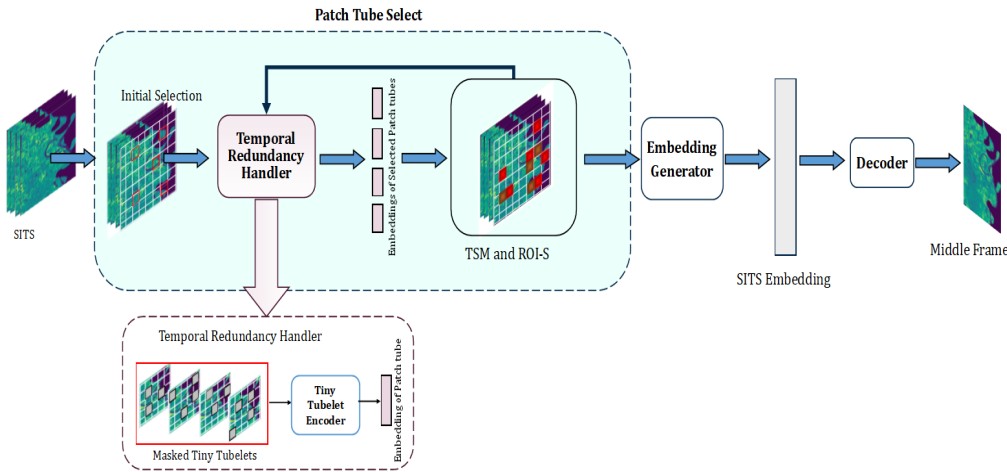

Figure 2: Model Architecture: SaTran

## 3.2 MODEL ARCHITECTURE

The architecture is given in Figure 2. PatchTubeSelect handles the spatiotemporal redundancy in SITS by selecting and processing the patch tubes using an attention mechanism. Temporal Redundancy Handler uses VideoMAE to handle the local temporal redundancy of the selected patch tubes. Both modules are described below:

**PatchTubeSelect:** A satellite image is gridded virtually into spatial patches. each patch consist of $n \times n$ pixels. A patch tube extends a patch in the temporal dimension. The patch tubes are passed to the Temporal Redundancy Handler which gives the representations of Patch tubes which are then forwarded to the Tube Selection Module (TSM) which is a sub-module in patch tube select. TSM utilizes attention scores to identify the top 'k' tubes, which are then passed to the Region of Interest Selector (ROI-S) sub-module. ROI-S determines the unprocessed neighboring tubes of the top 'k' tubes and generates a list of tubes to be processed in the subsequent iteration. Most of the neighboring patch tubes are excluded due to spatiotemporal redundancy and new patch tubes are randomly selected which then become the new regions of interest. This process iterates until a fraction ($1/x$) of the SITS is processed. The enhanced tube representations obtained from TSM are subsequently forwarded to the embedding generation module, which outputs the embedding of the entire SITS learned by the network across multiple iterations.

**TemporalRedundancyHandler:** It adapts VideoMAE for handling temporal redundancy. The patch tubes are divided into tiny tubelets of size ($T/t \times H/h \times W/w$) where $t$, $h$, $w$ depends on the satellite used and details are given in section 6.1. These tiny tubelets are randomly masked and the remaining tubelets are processed using vanilla Vision Transformer (ViT) Dosovitskiy et al. (2020). VideoMAE takes these tiny tubelets as inputs and uses the joint space-time cube embedding to obtain tubelet tokens. This reduces the volume of input data due to reduced spatial and temporal dimensions thus making the processing efficient. As the temporal redundancy in SITS persists for shorter lengths, the original high masking (95%) ratio in VideoMAE cannot be used for SITS. Also, there is less risk of information leakage in SITS, due to which we recommend reducing the masking ratio to 75% which is still significant and thus only a small number of tokens need to be processed by the encoder which further makes the process efficient. The comparative study for different masking ratios is given in the results section. To effectively capture high-level spatiotemporal details within these remaining tokens, we utilize the ViT backbone and incorporate joint space-time attention. This allows all pairs of tokens to interact with each other in the multi-head self-attention layer, enhancing the model's ability to understand complex spatiotemporal relationships.

**Embedding Generator:** It collects individual embeddings of all the processed patch tubes and uses a feedforward network to give joint representation of entire SITS. It consists of two linear layers.

**Decoder:** We pre-train SaTran for two tasks viz. reconstruction and classification (refer section 4) using different decoders one for each task.

The decoder for reconstruction task consists of trans-convolutional layers to perform the upsampling operations and construct the middle frame. Batch normalization is applied to ensure stability and efficiency in training. The decoder is trained using the mean squared error loss function for the model to generate the middle frame reconstructed images.

The decoder for the classification task consists of a simple feedforward network with two linear layers and a sigmoid activation for the output. We use binary cross-entropy loss as the loss function.

The decoders are used only in pre-training and are removed during fine-tuning.

# 4    PRE-TRAINING OF SATRAN

We optimized SaTran for two objectives in pre-training, given as follows:

**1) Reconstruction task:** We pre-train SatTran using middle frame reconstruction pretext task. The embeddings obtained from SatTran are passed to an additional decoder to construct the satellite image at the middle timestamp of the time series. The model learns to capture temporal dependencies and patterns inherent in the dynamic changes on Earth's surface. This pre-training task ensures that the model acquires a robust understanding of the temporal and spatial dynamics of SITS.

**2) Classification task:** The second pre-training task is a binary classification task to decide whether the given SITS is ordered or shuffled. SatTran learns whether the given input is temporally in the correct order or not. The model is given as input the original HSRSITS and the shuffled time series by randomly permuting the original time series. The loss function used is binary cross-entropy loss.

We separately pre-trained SatTran on two SITS datasets viz. MODIS (a moderate spatial resolution satellite) and Landsat-8 (high spatial resolution satellite) image time series, and named them SatTran-M and SatTran-L, respectively. This is due to the properties of the satellite data. The characteristics including height, width, bands, and time stamps (i.e. number of frames) are different for different satellites due to the different spatial, temporal, and spectral resolutions. Thus, the model pre-trained for one satellite data cannot be used for the other satellite data. This also requires optimization for suitable size of patch tubes and tiny tubelets for two satellite data. The details are given in the experimentation section 6.1.

We have also pre-trained VideoMAE for the reconstruction task in the same way as in the original paper for MODIS data for different masking ratios. The impact of different masking ratios by VideoMAE is given in the results section. VideoMAE did not work for Landsat-8 data because of the huge size of the images. So we pretrained VideoMAE-R, the resized version of VideoMAE given in section 6.1

**Pre-training data details:** SatTran is trained for 600 US counties on 7 years (2014-2020) of data. The length of time series taken in pre-training is 13 timestamps for MODIS and 7 timestamps for Landsat-8. The total number of instances used for pre-training is 1M for each satellite data and the size of the dataset is approx. 100GB for MODIS and 900GB for Landsat-8.

# 5    FINE-TUNING OF SATRAN

We have considered six earth observation applications as downstream tasks for testing SaTran. The applications include prediction of crop yield (CYP), snow cover (SCP), solar energy (SEP), soil moisture (SMP), cloud cover (CCP), and classification of land cover (LCC). The data used consists of SITS from two satellites MODIS and Landsat-8. We have taken around 2000 counties from different states of US for all applications considered. The ground truth data is specific to the application. The details of the study area, data used and its preprocessing steps are given in Appendix A. The application-specific details are given in Appendix B.2.

## 6 EXPERIMENTS AND RESULTS

**Experimental Setup:** All experiments are performed using Pytorch 1.11.0 and CUDA 11.7 on single A100 GPU server having 80GB GPU RAM. SaTran is pre-trained and fine-tuned using Adam optimizer for 35 and 15 epochs, respectively with a batch size of 8. The size of patch tubes for MODIS is $10 \times 10$ and for Landsat-8, it is $64 \times 64$. The size of tiny tubelets for MODIS is $5 \times 5 \times 2$ and for Landsat-8 it is $16 \times 16 \times 2$. All results are taken at batch size 8 until and unless stated.

We fine-tuned/trained models for data from years 2014-2017, data of 2018 is used for validation and we tested SaTran for two years (2019 and 2020) for all applications. All the predictions are done at timestamp $t$+1, for input time series having $t$ timestamps. If prediction needs to be done yearly, we averaged the predicted output for years 2019 and 2020. If application requires prediction at a monthly level (for SMP and SCP), average is taken for 24 predictions in two years, and so on. All application-specific details are mentioned in Appendix B.3 and B.4.

**Evaluation Metrics:** We use RMSE and MAE for prediction problems and accuracy & F1-score for classification.

### 6.1 MODELS FOR COMPARISON

We compare SaTran with existing models including two RGB video transformers (VideoMAE Tong et al. (2022), and ViViT Arnab et al. (2021)) and two SITS models - SITSFormer Yuan et al. (2022) and TSViT Tarasiou et al. (2023) developed for classification tasks (details given in Appendix C). VideoMAE is pretrained on the reconstruction task, whereas ViViT is not pretrained and is developed for the classification task. We adopted all four models for both prediction and classification downstream tasks. Also, none of these models is able to process Landsat-8 image time series at its original size and gives out-of-memory (OOM) error. We used two modification techniques in order to use these models for Landsat-8 data.

**1) Resize SITS:** We resize the Landsat-8 image time series to one-fourth along the spatial dimension keeping spectral and time dimensions same. If the original SITS is of size $B \times T \times H \times W$, the resized SITS is of size $B \times T \times \frac{H}{4} \times \frac{W}{4}$. The variants of the models mentioned above are represented by "$*$-R" in the results section e.g., VideoMAE is represented as VideoMAE-R.

**2) Segmentation:** We segment the original Landsat-8 image time series into 16 segments along the spatial dimension without tampering other two dimensions and each segment is of size $B \times T \times \frac{H}{4} \times \frac{W}{4}$. The existing models are applied to each segment and their embeddings are then concatenated together and passed through a feed-forward network to get the final predictions. This variant is represented by "$*$-S" e.g., VideoMAE is represented as VideoMAE-S.

### 6.2 RESULTS AND ANALYSIS

***Preliminary experiments: Pre-training using MODIS data***

We present preliminary experiments to pre-train SaTran and the existing model VideoMAE for the reconstruction task using MODIS data. The results are captured for different masking ratios. Table 1 presents GPU memory and time required for both models using batch size 8 for training. It can be observed from the table that SaTran takes approximately half of the GPU memory than that of VideoMAE. Though VideoMAE needs less time to execute for one epoch in comparison to SaTran, SaTran converges faster due to its attention mechanism of handling spatiotemporal redundancy and distributed approach of applying VideoMAE to patch tubes. Thus, total time taken by SaTran is much less than VideoMAE. Moreover, GPU memory requirements of VideoMAE increase exponentially by reducing the masking ratio. RMSE obtained by SaTran is lower than that of VideoMAE in all the cases. It can also be observed from the table that though best results are obtained for 60% masking for both models, change from 90% to 75% is more than the change from 75% to 60%. However, computational requirements for 60% are more than 75%. Thus we recommend 75% masking of tiny tubelets this is unlike what is recommended by VideoMAE (95%) for RGB videos. VideoMAE suggests high masking ratio to avoid data leaks. In our case data leaks are avoided by not processing the redundant patches and 75% masking is sufficient to avoid data leaks in non-redundant patches. We also conducted experiments for batch size 16 where VideoMAE gives out-of-memory error when masking ratio is reduced to 60%. The results are given in Appendix D.1.

Table 1: Memory and Time requirements for SaTran and VideoMAE in pretraining on SITS for Reconstruction task using batch size 8. VideoMAE-R uses reduced size of Landsat-8 SITS.

| | MODIS Data | | | | | | | |
| | GPU memory | | Time per epoch (in hours) | | Total time (in hours) | | RMSE | |
| Masking Ratio | VideoMAE | SaTran | VideoMAE | SaTran | VideoMAE | SaTran | VideoMAE | SaTran |
|---|---|---|---|---|---|---|---|---|
| 90 | 30GB | 20GB | 5.815 | 5.833 | 232.612 | 174.999 | 0.2052 | 0.1879 |
| 75 | 39GB | 21GB | 6.293 | 6.645 | 251.724 | 193.500 | 0.1984 | 0.1856 |
| 60 | 49GB | 22GB | 6.544 | 6.450 | 261.776 | 199.374 | 0.1902 | 0.1848 |
| | Landsat-8 Data | | | | | | | |
| Masking Ratio | VideoMAE-R | SaTran | VideoMAE-R | SaTran | VideoMAE-R | SaTran | VideoMAE | SaTran |
| 90 | 52GB | 60GB | 5.925 | 7.259 | 237.000 | 217.794 | 0.2622 | 0.1927 |
| 75 | 58GB | 65GB | 6.715 | 7.864 | 268.600 | 235.926 | 0.2612 | 0.1901 |
| 60 | 64GB | 72GB | 6.920 | 8.764 | 276.800 | 262.935 | 0.2606 | 0.1893 |

Table 2: Comparison of SaTran with existing transformer and SITS models for Landsat-8 data for different earth observation applications for prediction and classification using RMSE and F1-score, respectively. VideoMAE, ViViT, SITSFormer, and TSViT, all the models through OOM error while using original resolution of Landsat-8 image time series

| | Prediction Problems (RMSE) | | | | | Classification (LCC) | | |
| Model | CYP | SEP | SCP | SMP | CCP | Accuracy | Precision | F1-score |
|---|---|---|---|---|---|---|---|---|
| VideoMAE-R (90%) Tong et al. (2022) | 9.2038 | 6.7957 | 22.2991 | 7.5549 | 18.6554 | 0.6823 | 0.4589 | 0.4397 |
| VideoMAE-R (75%) Tong et al. (2022) | 7.3457 | 6.3511 | 19.1947 | 6.7957 | 18.8426 | 0.7101 | 0.4980 | 0.4605 |
| SITSFormer-R Yuan et al. (2022) | 6.9860 | 6.8325 | 19.8670 | 5.9716 | 16.2534 | 0.8053 | 0.5763 | 0.5874 |
| ViViT-R Arnab et al. (2021) | 8.7552 | 6.4307 | 22.0058 | 6.3987 | 18.3996 | 0.6915 | 0.5694 | 0.5201 |
| TSViT-R Tarasiou et al. (2023) | 9.3365 | 7.1937 | 19.8471 | 4.7591 | 17.8773 | 0.8368 | 0.6923 | 0.6186 |
| VideoMAE-S (90%) Tong et al. (2022) | 8.6085 | 6.1708 | 21.4295 | 7.1644 | 18.3846 | 0.7462 | 0.5031 | 0.4703 |
| VideoMAE-S (75%) Tong et al. (2022) | 7.3457 | 5.9611 | 17.5016 | 6.4978 | 17.7153 | 0.7551 | 0.5112 | 0.4957 |
| SITSFormer-S Yuan et al. (2022) | 6.6661 | 6.4081 | 19.7654 | 5.4121 | 15.3935 | 0.8289 | 0.6041 | 0.6153 |
| ViViT-S Arnab et al. (2021) | 7.1626 | 6.0159 | 21.6651 | 6.0159 | 16.7468 | 0.7263 | 0.6198 | 0.5562 |
| TSViT-S Tarasiou et al. (2023) | 8.3497 | 6.1563 | 17.2997 | 4.4878 | 17.0686 | 0.8506 | 0.7212 | 0.6415 |
| **SaTran (75%) (our)** | **5.5584** | **5.0193** | **15.3112** | **3.0775** | **12.0716** | **0.9486** | **0.8172** | **0.6967** |
| Mean ± Std. dev. | 5.6170 ± 0.067 | 5.1196 ± 0.07 | 15.4473 ± 0.137 | 3.1299 ± 0.039 | 12.1601 ± 0.088 | 0.9231 ± 0.0255 | 0.7845 ± 0.0327 | 0.6571 ± 0.0396 |

### Pre-training using Landsat-8 data

The memory and time requirements for models pretrained using Landsat-8 data are given in Table 1. We use VideoMAE-R variant for Landsat-8 data because Landsat-8 is a high spatial resolution satellite and its image time series is huge and cannot be processed using VideoMAE even for masking ratio of 90% on systems with the specifications mentioned in section 6. Whereas, SaTran is successfully able to process Landsat-8 image time series in its original resolution and also requires less time as compared to VideoMAE but the GPU memory used is slightly more than that for VideoMAE-R.

### Ablation study:

We performed ablation study for deciding traversal fraction (1/x) of SITS, batch size, masking ratio for tiny tubelets, and pretraining tasks. The details are given in Appendix D. We fixed masking ratio for the experiments at 75%. We concluded empirically that optimal traversal for MODIS $x=4$ and for Landsat $x=3$, respectively (refer Figure 3 in Appendix D.2). We used these selected values of all parameters in all the experiments.

### Comparison of SaTran with existing models:

We executed all experiments five times and reported the best case scenarios in the paper. Here, we compare SaTran with existing models (given in section 6.1) for various downstream tasks using Landsat-8 data. Table 2 represents the RMSE obtained by different models for various downstream tasks Corresponding MAE values are given in Table 9 in Appendix E. It can be observed from the table that none of the existing models were able to process the Landsat-8 time series with its original dimensions due to its large size. All the existing models suffer from out-of-memory (OOM) error. The *resize* and *segmentation* variants of models are able to process Landsat-8 SITS, but their performance is inferior to that of SaTran. SITSFormer and TSViT perform better for classification problems as they were originally developed for similar classification problems and we adapted them for prediction tasks. The error obtained by the *resize* variant of the models is larger than their corresponding *segmentation* variants. This is because the spatial resolution of SITS is degraded.

Table 3: Comparison of SaTran with competitive models for memory & time requirements for CYP

| Model | # training parameters | Training time (in hours) | GPU memory |
|---|---|---|---|
| VideoMAE-R (90%) Tong et al. (2022) | 431M | 14.5 | 48GB |
| VideoMAE-R (75%) Tong et al. (2022) | 431M | 16.0 | 54GB |
| SITSFormer-R Yuan et al. (2022) | 500M | 18.0 | 42GB |
| ViViT-R Arnab et al. (2021) | 86M | 9.16 | 52GB |
| TSViT-R Tarasiou et al. (2023) | 360M | 12.5 | 56GB |
| VideoMAE-S (90%) Tong et al. (2022) | 451M | 27.5 | 67GB |
| VideoMAE-S(75%) Tong et al. (2022) | 451M | 29.1 | 72GB |
| SITSFormer-S Yuan et al. (2022) | 553M | 48.0 | 58GB |
| ViViT-S Arnab et al. (2021) | 92M | 15.3 | 72GB |
| TSViT-S Tarasiou et al. (2023) | 490M | 22.2 | 66GB |
| SaTran (75%) | 311M | 16.4 | 64GB |

In a few cases, the performance of *resize* variants degrades even from MODIS data (see Table 8 in Appendix E) because the temporal resolution of Landsat-8 is already coarser than MODIS and due to resizing the spatial resolution is also compromised, thus leading to further loss of information. On the other hand, the proposed model SaTran not only successfully processes the Landsat-8 image time series at its original resolution but also outperforms both the variants of all existing models for all the tasks.

***Memory and time requirements of the Models***

The GPU memory, time, and number of training parameters are given in Table 3. It can be observed from the table that only ViViT has lesser number of training parameters than that of SaTran, and all other models (both variants) need more training parameters than SaTran. Also, the training time of SaTran is comparable to *resize* variant of existing models and it is lesser than the *segmentation* variants which use the same resolution of SITS as that used in SaTran. It can also be observed that the GPU memory requirements of SaTran are also comparable to the competing models.

It is evident from Table 2 and 3 that SaTran outperforms all the existing models and has reasonable time and space requirements. None of the baselines either processes the Landsat image time series at coarser spatial resolution (*resize* variant) or by segmenting in the spatial dimension performs well in solving the earth observation applications. Thus, this establishes the requirement of SaTran which can efficiently process large size SITS to give learned representations and can be used for various applications.

We also conducted experiments for MODIS image time series for fair comparison because existing models are able to process MODIS data in its original spatial resolution. Due to space constraints, results are given in Appendix E. The results show that SaTran outperforms all the existing models.

**Limitations of the model**

The proposed transformer can work for SITS obtained from any satellite irrespective of its spatial, temporal, and spectral resolutions. The transformer can be adopted for any SITS by adjusting a few hyper-parameters like patch size, traversal fraction, learning rate, etc. We performed experiments only for MODIS and Landsat-8, but it can be used for other satellites like Sentinel-2.

# 7 CONCLUSION

We presented a large SITS model, SaTran, for large size SITS trained in self-supervised learning setup using two-fold data redundancy handling. SaTran introduces two novel designs - automatic patch tube selection and a distributed approach of applying tube masking on tiny tubelets. SaTran reduces the memory requirements by approximately a factor of 2. Experimental results show that due to short temporal redundancy, it is not recommended to have a very high masking ratio to achieve better results. Our experiments also demonstrate that the time taken by SaTran increases sublinearly with an increase in image size e.g., we observed an increase of 18% in processing time for 900GB of Landsat-8 data in comparison to 100GB of moderate resolution MODIS data. We have fine-tuned SaTran for various downstream tasks and compared the performance with existing video vision models and existing SITS models. The existing models are unable to process large-size Landsat-8 image time series in its original spatial resolution. It is evident from the results that SaTran efficiently processes the large-size SITS and outperforms all the competitive models for all downstream tasks.

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

# 8 SUPPLEMENTARY MATERIAL

## A APPENDIX: DATASET DETAILS

### A.1 DATASET USED

For MODIS, time series for one year is $200 \times 250 \times 5(\#bands) \times 46(\#timestamps)$ and for Landsat-8, it is $2000 \times 2500 \times 5 (\#bands) \times 23 (\#timestamps)$. For a fair comparison, we have used data for the same years (2014-2020) for both satellites and used only the bands common to both satellites i.e. Red, Green, Blue, NIR, SWIR.

### A.2 DATA PREPROCESSING

Satellite data is captured as raw multispectral images and needs pre-processing before it can be used for pertaining or any downstream task. The data preparation steps are given below:

1. *Handling cloudy pixels:* We only consider the scenes having cloud cover of less than 15 percentile at every pixel. It is taken care of by using the simple composite algorithm of the Google Earth engine goo.

2. *Handling missing values:* Satellite data often suffers missing data due to clouds on the day of capture, technical failures, and other reasons. We estimate missing data at pixel level using nearest neighbor interpolation and for timestamp level we use linear interpolation paul (2019).

3. *Bits Precision:* Landsat-8 images have float values at every pixel. Float values require 32 bits to store a single pixel which leads to the requirement of large storage space for Landsat images. Moreover working with earth observation applications, analysis of historical data is important which increases the number of images to be processed. To save storage space we used the bits precision compression technique Hubara et al. (2016) and converted all the float values to unsigned integer values as done in Goyal et al. (2024).

## B APPENDIX: APPLICATION SPECIFIC DETAILS

### B.1 SIGNIFICANCE OF DIFFERENT EARTH OBSERVATION APPLICATIONS CONSIDERED

Proper modeling of Earth Observation Applications is critically important for developing acceptable and deployable solutions.

- *Crop Yield Prediction*: Accurate and timely crop yield forecasting plays a crucial role in supporting governments and farmers in making informed decisions and planning strategies related to agricultural production. Governments rely on crop yield prediction to formulate policies regarding import and export, determine pricing mechanisms, and address potential threats to food security. By having access to reliable crop yield predictions, governments can effectively manage resources, allocate budgets, and implement measures to ensure stability and sustainability in the agricultural sector. For individual farmers, crop yield prediction provides valuable insights into the expected output of their crops. This information enables them to plan their farming activities, make informed choices regarding inputs such as seeds, fertilizers, and pesticides, and manage their resources effectively. Farmers can adjust their cultivation practices, adopt appropriate risk management strategies, and optimize their harvest and post-harvest activities based on the anticipated yield.

- *Soil Moisture Prediction*: Soil moisture prediction indirectly impacts many other earth observation applications. It helps in studying long-term climate patterns, managing water resources in any area, drought management, and allocation of water resources for agriculture and urban use. It helps the infrastructure industry to decide on the design and material for bridges and pipelines for their stability and longevity. It even helps in disaster preparation by identifying high-risk zones for landslides or droughts. Soil moisture analysis helps environmental planners to implement erosion control measures effectively as less moisture in the soil leads to more soil erosion in the area.

- *Solar Energy Prediction*: Solar energy prediction is done to find a suitable location for the installation of solar plants and reduce the dependence on fossil fuels for economic development. It helps in planning resource utilization to meet energy demands. The predictions play a vital role in shaping the future of energy production and consumption, leading the way towards a more sustainable and eco-friendly energy source.

- *Snow Cover Prediction*: Snow cover plays a vital role in agriculture as it provides moisture to the soil during spring melt. Snow cover prediction is crucial for managing reservoirs, planning irrigation, road maintenance, ensuring safer transportation, and preventing flooding. Also, predicting the snow cover can help in identifying high-risk areas to issue early warnings. It also helps in planning tourism for a nation during the winter season. Predicting snow cover can be carried out using drones, sensors, or by manually visiting the place. However, snow cover prediction with the help of satellite data is useful in getting an estimate of the presence of snow cover in dangerous and inaccessible areas.

- *Cloud Cover Prediction*: Cloud cover prediction has a significant impact on the tourism industry and Maritime Operations of a nation. Tourism agencies and travelers consider historical cloud cover patterns when planning vacations and destinations. Mariners use cloud cover predictions, especially in coastal and navigational areas, to plan shipping routes. Cloud cover can affect visibility and safe navigation at sea. Cloud cover also helps in air quality monitoring as cloud cover affects the dispersion of pollutants in the atmosphere. Air quality researchers use cloud cover predictions to understand how pollutants disperse and accumulate, aiding in air quality monitoring and management. Hydrologists and water resource managers use cloud cover predictions to model evaporation rates from water bodies. Cloud cover influences the amount of solar radiation reaching the surface, affecting evaporation rates and water availability.

There are a couple of other applications like semantic segmentation and object detection which are important in satellite image technology. However, these applications are done using benchmark datasets which consists of images covering less area in terms of $km^2$ and thus are of small size. These applications are thus out of scope of our paper and we have not considered them.

## B.2 DATA USED

- Crop Yield Prediction (CYP): The crop yield data for U.S. counties utilized in this study is taken from Quick Stat, a comprehensive database compiled by the United States Department of Agriculture (USDA) USDA (2010). The data used in this study spans the period from 2002 to 2020. The yield values are quantified in bushels per acre (bu/ac), offering a standardized unit for assessing and comparing crop productivity across different regions.

Table 4: Length of time series used for different downstream applications

| Satellite | CYP | SMP | SCP | SEP | CCP | LCC |
|-----------|-----|-----|-----|-----|-----|-----|
| MODIS     | 32  | 12  | 12  | 24  | 24  | 46  |
| Landsat-8 | 16  | 6   | 6   | 12  | 12  | 23  |

We used top producer counties of soybean from states - Michigan, North Dakota, Arkansas, Indiana, Tennessee, Ohio, South Dakota, Iowa, Kansas, Kentucky, Minnesota, Mississippi, Missouri, Nebraska, Illinois, and Wisconsin.

- Snow Cover Prediction (SCP): The snow cover data has been acquired at the county level using the MODIS product MOD10A1. This dataset offers information about snow cover extent with a spatial resolution of 500 meters using normalized difference snow index (NDSI). NDSI is a key indicator, representing the percentage of the area covered by snow. We consider 300 counties from states experiencing annual snowfall of more than 250 inches. The counties lie in the states of New Hemisphere, Washington, Oregon, California, Colorado, and Utah.

- Soil Moisture Prediction (SMP): The soil moisture data is acquired at the county level for every month using NASA-USDA Enhanced SMAP Global soil moisture data NASA-USDA. This dataset was developed by the Hydrological Science Laboratory at NASA's Goddard Space Flight Center in cooperation with USDA Foreign Agricultural Services and USDA Hydrology and Remote Sensing Lab. The Soil Moisture Active Passive (SMAP) instrument measures the amount of water in the surface soil everywhere on Earth. The value represents the amount of water in mm. The data is available from 2016 and thus data used in this study is for 5 years from 2016 to 2020. We consider 275 counties from the states- Iowa, Kansas, Illinois, Kentucky, and Indiana.

- Solar Energy Prediction (SEP): The information about solar energy produced in a county has been acquired from visual crossing Visual Crossing Corporation (2019). The value represents the total solar energy produced in MJ/m2 for a county in a day. The data used in this work spans the period from 2014 to 2020. The counties considered lie in the states- Iowa, Kansas, Illinois, Kentucky, and Indiana.

- Cloud Cover Prediction (CCP): Cloud cover represents the proportion of the sky covered by clouds throughout the day. We acquired data captured on a daily basis from visual crossing Visual Crossing Corporation (2019). The data used in this study spans the period from 2014 to 2020. The counties considered lie in the states- Iowa, Kansas, Illinois, Kentucky, and Indiana.

- Land Cover Classification (LCC): Land cover can be classified into a different number of classes. We used MCD12Q1 modis which classifies land cover into 17 classes. The dataset provides annotations for the land cover at a spatial resolution of 500m and yearly granularity. The data used in this study spans the period from 2014 to 2020. The counties considered belong to states- Iowa, Kansas, Illinois, Kentucky, and Indiana.

### B.3 Deciding length of time series:

The length of time series for an instance for every application is different due to the difference in the visiting frequency of satellite systems. For e.g. in snow cover and solar energy prediction the time series considered is from the last three months and the prediction is for the next month for snow cover and the next fortnight for solar energy. The details are given in Table 4.

### B.4 Learning Rate used

The learning rate $\eta$ for the optimizer is decided while tuning hyper-parameters for each model and application. For CYP, CCP, SMP, SEP, SCP, LCC $\eta$= [0.0003, 0.001, 0.001, 0.0001, 0.00001, 0.0001], respectively.

Table 5: Memory and Time required for MODIS with batch size 16

| Masking | GPU memory | | Time per epoch (in hours) | | Total time (in hours) | | RMSE | |
|---|---|---|---|---|---|---|---|---|
| | VideoMAE | SaTran | VideoMAE | SaTran | VideoMAE | SaTran | VideoMAE | SaTran |
| 90 | 54 GB | 30GB | 3.5472 | 3.6111 | 177.3600 | 126.3891 | 0.2081 | 0.1874 |
| 75 | 70GB | 31GB | 3.6694 | 3.7222 | 183.4700 | 130.2775 | 0.1975 | 0.1861 |
| 60 | OOM | 32GB | OOM | 4.0292 | OOM | 141.0220 | OOM | 0.1818 |

## C APPENDIX: MODELS FOR COMPARISON

We compared SaTran with two RGB video transformers and two SITS models. These models are incapable of processing Landsat-8 image time series in its original spatial resolution. So we have used their *resize* and *segmentation* variants described in section 6.1. The details of the models used for comparison are as follows:

1. **VideoMAE Tong et al. (2022):** VideoMAE divides the RGB videos into tubelets and uses a high masking ratio (95%) to handle temporal redundancy in the videos. However, VideoMAE pre-trained on RGB videos is not suitable for SITS data. Thus, for a fair comparison, we tried to pre-train VideoMAE for the Landsat-8 image time series. We are not able to do so for Landsat-8 data in its original fine resolution due to huge size of images and the GPU was out of memory even after reducing batch size to 2. Thus, we pretrain the *resize* variant of the model mentioned in section 6.1 using 90% and 75% masking ratio and used this variant for all the downstream tasks.

2. **ViViT Arnab et al. (2021):** ViViTArnab et al. (2021) is originally developed for RGB videos. Image time series of dimension $T \times H \times W$ is converted into multiple tubelets of dimension $z \times h \times w$ and tokens are extracted from all three dimensions. These tubelet embeddings efficiently capture spatiotemporal information by representing non-overlapping spatiotemporal patches. We adapted it for SITS data and used the factorized encoder version of the transformer.

3. **SITSFormer Yuan et al. (2022):** SITSFormer uses the neighboring pixels and then predict the label of the center pixel using the BERT model. SITSFormer is originally designed for classification tasks. We have adapted it for prediction tasks by replacing the classification head with two linear layers and an output layer for prediction.

4. **TSViT Tarasiou et al. (2023):** TSViT is also designed for classification tasks. It splits a SITS instance into non-overlapping patches in space and time which are tokenized and processed by a factorized temporal-spatial encoder. It uses class-specific *cls* tokens as inductive bias to improve the model prformance. We have adapted it for prediction tasks by replacing the classification head with prediction head consisting of two linear layers and an output layer for prediction.

## D APPENDIX: ABLATION STUDY

### D.1 RESULTS OF MODIS FOR BATCH SIZE 16

We also conducted experiments for different masking ratio and batch size while training. When batch size is increased to 16, VideoMAE is not able to process MODIS time series also if the masking ratio is less than 75%. However, SaTran performs well even at a lower masking ratio as well. Thus, for fair comparison, we performed all results at 8 batch size. Also, this shows that SaTran is an efficient model in terms of computation requirements in comparison to VideoMAE for SITS analytics. The results are given in Table 5.

### D.2 SELECTING OPTIMAL $x$ FOR PARTIAL TRAVERSAL OF SITS

Figure 3 shows the computation time required and error curve for the reconstruction task by varying $x$ as 5,4,3, and 2. The masking ratio is fixed at 75%. It can be observed from the figure that, there is an improvement of $\approx 6\%$ in the performance of the SaTran-M when we traverse $1/x$ of the time series $x$ changing $x$ from 5 to 4, and the error did not change much after that. Similarly, for SaTran-L,

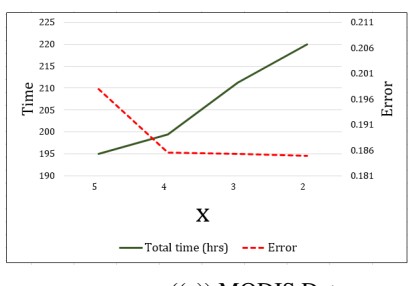 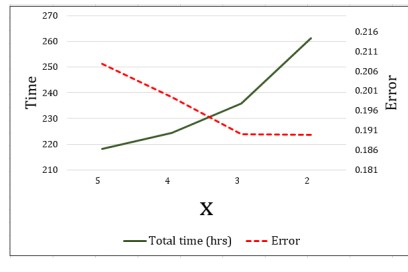

((a)) MODIS Data                    ((b)) Landsat-8 Data

Figure 3: Deciding x for (1/x)th traversal of SITS

Table 6: Impact of masking ratios on performance of SaTran and VideoMAE (pre-trained on Reconstruction Task using MODIS data) for various applications (RMSE)

| Application | 90% masking | | 75% masking | | 60% masking | |
|---|---|---|---|---|---|---|
| | **VideoMAE** | **SaTran** | **VideoMAE** | **SaTran** | **VideoMAE** | **SaTran** |
| CYP | 8.1661 | 7.7240 | 7.9826 | 7.3894 | 7.5549 | 7.3862 |
| SCP | 21.175 | 18.175 | 20.746 | 18.115 | 20.449 | 18.057 |
| SEP | 6.4693 | 6.3361 | 6.3983 | 6.1997 | 6.2302 | 6.1806 |
| CCP | 14.751 | 14.342 | 14.732 | 13.810 | 14.324 | 13.764 |
| SMP | 6.5409 | 3.8088 | 5.1252 | 3.5338 | 4.8953 | 3.5131 |

the maximum improvement is observed when $x$ changes from 4 to 3. The computation time required from $x = 5$ to 4 for SaTran-M and from $x = 4$ to 3 does not increase much, but it increases linearly after that. The results for the downstream tasks are given for the optimal traversal in both cases (For MODIS $x=4$ and for Landsat $x=3$).

### D.3 Impact of different masking ratios on downstream applications

To analyze the impact of different masking ratios of tiny tubelets, we performed experiments for prediction downstream applications using MODIS data. The table 6 gives RMSE for various applications. It can be observed from Table that SaTran outperforms VideoMAE for all the applications for all masking ratios. For example, when using 75% masking, RMSE reduces by approximately 8%, 14.5%, 3%, 6.6%, and 45% for soybean, snow cover, solar energy, cloud cover, and soil moisture prediction, respectively.

### D.4 Impact of pretraining tasks on SaTran

Table 7 gives RMSE values and shows the impact of training SaTran for different pre-training tasks. The results in table are presented with 1/4 traversal of MODIS image time series and 75% masking of the tubelets. It can be observed that RMSE reduces by a significant margin for all the applications when SaTran is pre-trained further using classification task. In case of MODIS data, maximum improvement is seen in crop yield prediction with approximately 12% reduction in RMSE, followed by 11% in cloud cover prediction. The minimum improvement observed is 4% for soil moisture prediction.

Similarly, Table 7 presents the RMSE obtained by SaTran when pre-trained only for the reconstruction task and both reconstruction and classification tasks using high spatial resolution Landsat-8 image time series. The results in table are presented with 1/3 traversal of SITS and 75% masking of the tubelets for Landsat-8. It can be observed from the table that RMSE reduces when SaTran is pre-trained for both tasks. The maximum improvement is observed in soil moisture prediction with a reduction in RMSE by 13%.

Table 7: Impact of different pretraining tasks on performance of SaTran for various downstream applications

| MODIS Data | | | |
|---|---|---|---|
| Application | Pre-trained on reconstruction task | Pre-trained on reconstruction + classification tasks | % improvement |
| CYP | 7.3894 | 6.5825 | 12.25 |
| SCP | 18.115 | 16.847 | 7.520 |
| SEP | 6.1997 | 5.7992 | 6.900 |
| CCP | 13.810 | 12.345 | 11.86 |
| SMP | 3.5338 | 3.3842 | 4.420 |
| Landsat-8 Data | | | |
| Application | Pre-trained on reconstruction task | Pre-trained on reconstruction + classification tasks | % improvement |
| CYP | 5.9046 | 5.5584 | 6.220 |
| SCP | 15.641 | 15.311 | 2.150 |
| SEP | 5.2523 | 5.0193 | 4.640 |
| CCP | 12.195 | 12.071 | 1.020 |
| SMP | 3.4844 | 3.0775 | 13.22 |

Table 8: Comparison of SaTran with competitive models for various downstream applications using MODIS data (RMSE)

| Model | CYP | SEP | SCP | SMP | CCP |
|---|---|---|---|---|---|
| VideoMAE (90%) Tong et al. (2022) | 8.1661 | 6.4693 | 21.1754 | 6.5409 | 14.7517 |
| VideoMAE (75%) Tong et al. (2022) | 7.9826 | 6.3983 | 20.7469 | 5.1252 | 14.7325 |
| SITSFormerYuan et al. (2022) | 8.2761 | 7.5721 | 19.9875 | 6.1574 | 19.2559 |
| ViViT Arnab et al. (2021) | 9.2437 | 6.7068 | 20.0251 | 6.4843 | 17.2785 |
| TSViT Tarasiou et al. (2023) | 9.1051 | 6.2486 | 17.4759 | 4.9115 | 18.6722 |
| **SaTran (75%) (our)** | **6.5825** | **5.7992** | **16.8472** | **3.3842** | **12.3456** |

# E    APPENDIX: ADDITIONAL RESULTS OF COMPARISON ON MODIS TIME SERIES

We have also compared the proposed model SaTran for MODIS image time series. Although, this is not a high spatial resolution image time series but, the existing models can only process MODIS time series at its original spatial resolution. Table 8 shows the RMSE obtained by different models using MODIS image time series. Similarly, Table 9 shows the MAE for different models for various applications.It is clearly evident from the tables that the proposed model outperforms all the existing models.

Table 9: Comparison of SaTran with competitive models for various downstream applications using MODIS data (all values in MAE)

| Model | CYP | SEP | SCP | SMP | CCP |
|---|---|---|---|---|---|
| VideoMAE (90%) Tong et al. (2022) | 7.5642 | 5.8412 | 19.9263 | 6.0165 | 13.5384 |
| VideoMAE (75%) Tong et al. (2022) | 6.7193 | 5.8326 | 19.4203 | 5.9671 | 13.0047 |
| SITSFormerYuan et al. (2022) | 7.6921 | 6.4369 | 17.1149 | 5.3678 | 17.8264 |
| ViViT Arnab et al. (2021) | 8.0801 | 5.6912 | 18.3468 | 5.7002 | 15.8643 |
| TSViT Tarasiou et al. (2023) | 7.6194 | 5.5661 | 15.7339 | 4.4764 | 16.8339 |
| **SaTran (75%) (our)** | **6.4866** | **5.5696** | **15.6126** | **3.8660** | **11.9826** |

