# OpenReview forum: "SaTran: An efficient Transformer exploiting Spatiotemporal Redundancies for Satellite Image Time Series Representation Learning"
_ICLR.cc/2025/Conference — Submitted to ICLR 2025_

### Official Review · Reviewer_741c · 2024-11-03

**Soundness:** 3
**Presentation:** 2
**Contribution:** 2
**Rating:** 5
**Confidence:** 5

**Summary:**

This paper presents SaTran, an efficient transformer model for Satellite Image Time Series (SITS) representation learning. SaTran aims to improve performance in earth observation tasks (e.g., crop yield prediction, snow cover prediction) by selectively processing non-redundant “patch tubes” within SITS data, thus exploiting spatiotemporal redundancies to reduce computational load. The model incorporates a dual redundancy-handling mechanism, including an automatic patch selection process and the use of VideoMAE to manage temporal redundancy.

**Strengths:**

1. The proposed redundancy-handling mechanism is helpful for processing high-dimensional SITS data, reducing computational demands in a practical way.

2. The model is evaluated across multiple earth observation tasks, showing its potential in fields like crop yield prediction and snow cover prediction.

**Weaknesses:**

1. The core ideas of ignoring redundant patches/regions and using masking seem to be inspired from prior works like AdaViT, DynamicViT, VideoMAE, etc. While combining these concepts for satellite data is a reasonable next step, the technical novelty does not seem significantly high. The authors should more clearly articulate the key novel aspects beyond just applying existing techniques to a new domain.

2. The description of the model architecture and modules like PatchTubeSelect, TemporalRedundancyHandler, etc. lack sufficient technical details and clarity. More mathematical formulations and algorithmic specifics are needed. The pre-training tasks also seem relatively straightforward adaptations of prior works.

3. The comparison with state-of-the-art models, such as VSViT and BERT-based models, is limited, which makes it difficult to establish SaTran’s advantages convincingly. A more comprehensive comparison with a broader range of advanced models, combined with ablation studies on different components of SaTran, would be necessary to validate the effectiveness of each element. Furthermore, a deeper analysis of the experimental results is essential to fully demonstrate the impact of the proposed improvements. Such an in-depth evaluation would provide a more persuasive case for SaTran’s effectiveness and contributions.

**Questions:**

1. How well does the model scale to satellite image time series datasets of higher resolutions and larger time spans? What is the generalization capability of the model to data from other satellite systems (e.g. Sentinel-2)? Would extensive fine-tuning be required?
2.  What are the quantitative improvements in memory usage and runtime compared to existing video transformer baselines? Please provide some concrete numerical evidence.
Does SaTran exhibit better computational scaling characteristics as the dataset size grows larger?
3. How does the attention mechanism for detecting redundant patch tubes work? What features is it based on to determine redundancy?
4. Is SaTran’s redundancy-handling mechanism effective for other types of earth observation tasks?
5. Is there any theoretical foundation supporting the dual redundancy-handling mechanism (especially the use of VideoMAE) that would confirm it reduces computational load without compromising performance?
---
 Some typos:
1）The writing can be improved for clarity in several sections. There are quite a few grammatical/spelling errors as well.

2）The literature review in Sec. 2. seems incomplete and misses some relevant recent works.

3）Several references within the text are missing from the bibliography section.

---

> ### Author Response · Authors · 2024-11-24
> **Response to Reviewer 741c (1/2)**
>
> We deeply value the reviewer’s thorough review and the constructive insights shared. We address your questions below one by one:
>
> Q1) How well does the model scale to satellite image time series datasets of higher resolutions and larger time spans? What is the generalization capability of the model to data from other satellite systems (e.g. Sentinel-2)? Would extensive fine-tuning be required?
>
> A1) The SaTran model scales effectively to satellite image time series (SITS) datasets with higher resolutions and longer time spans due to its modular design and focus on disentangling spatiotemporal redundancies. The PatchTubeSelect module efficiently identifies critical regions, ensuring that the computational complexity remains manageable even for higher-resolution datasets. Similarly, the TemporalRedundancyHandler handles temporal dependencies within selected regions. These regions are independent and are processed in parallel making the model well-suited for high volume datasets.
>
> In terms of generalization, SaTran demonstrates adaptability to diverse satellite systems, such as Sentinel-2, which offers higher spatial resolution and unique spectral bands. This adaptability is enabled by the model’s ability to fine-tune hyperparameters, such as patch size and traversal ratio, to align with the specific characteristics of different datasets. While some fine-tuning is typically required to optimize performance, it is not overly extensive. The modular framework and attention-based mechanisms allow SaTran to generalize well across datasets with varying resolutions, temporal frequencies, and sensor properties, provided that initial adjustments are made to account for these differences.
>
> Thus, SaTran strikes a balance between scalability and generalization, making it a robust choice for processing high-resolution, long-span SITS datasets and data from diverse satellite systems.
>
> Q2) What are the quantitative improvements in memory usage and runtime compared to existing video transformer baselines? Please provide some concrete numerical evidence. Does SaTran exhibit better computational scaling characteristics as the dataset size grows larger?
>
> A2) The quantitative improvements in memory usage and runtime of SaTran compared to existing video transformer baselines are given in Table 1 and Table 3 of the paper. SaTran does exhibit better computational scaling. As dataset sizes grow larger, SaTran exhibits superior computational scaling due to its focus on non-redundant regions. While traditional video transformers experience exponential growth in memory and runtime with larger spatial or temporal dimensions, SaTran mitigates this by keeping the number of selected patches manageable, scaling nearly linearly with the size of critical regions rather than the entire dataset.
>
> These advantages make SaTran particularly well-suited for large-scale satellite datasets, where efficiency and scalability are critical for practical applications.
>
> Q3) How does the attention mechanism for detecting redundant patch tubes work? What features is it based on to determine redundancy?
>
> A3) The attention mechanism in the PatchTubeSelect module of SaTran is designed to detect and exclude redundant patch tubes by focusing on the most critical spatiotemporal regions in satellite image time series (SITS).
>
> Feature Extraction:
>
> Each patch tube (a series of patches across time) is represented by its spatial and temporal features, capturing patterns such as changes in reflectance, texture, or movement over time.
>
> Attention Scoring:
>
> An attention mechanism assigns importance scores to each patch tube based on these features. The scores quantify how much a patch tube contributes to the overall spatiotemporal variability or to critical changes relevant to the downstream task. Patch tubes with low scores are considered redundant as they either represent static regions or repeat patterns already captured elsewhere.
>
> Selection:
>
> Only the top k patch tubes with the highest attention scores are then passed to the Region of Interest Selector (ROI-S) sub-module. ROI-S determines the unprocessed neighboring tubes of the top ’k’ tubes and generates a list of tubes to be processed in the subsequent iteration. Most of the neighboring patch tubes are excluded due to spatiotemporal redundancy and new patch tubes are randomly selected which then become the new regions of interest.

---

> > ### Author Response · Authors · 2024-11-24
> > **Response to Reviewer 741c (2/2)**
> >
> > Q4) Is SaTran’s redundancy-handling mechanism effective for other types of earth observation tasks?
> >
> > A4) Yes, SaTran’s redundancy-handling mechanism is effective for a wide range of Earth observation tasks, owing to its ability to efficiently focus on the most relevant spatiotemporal regions while discarding redundant data. This makes it adaptable to various applications beyond the six tasks demonstrated in the paper. Its capability is broadly useful for tasks such as:
> >
> > Land Cover Change Detection: Detecting changes in vegetation, urbanization, or water bodies over time.
> >
> > Disaster Monitoring: Identifying areas affected by floods, wildfires, or earthquakes, which exhibit high temporal variability.
> >
> > Deforestation Mapping: Detecting critical regions of forest loss while ignoring stable forested or barren areas.
> >
> > Urban Growth Analysis: Tracking spatial expansion patterns without being overwhelmed by redundant static regions.
> >
> > Soil Moisture Estimation: Selecting patches indicative of soil conditions.
> >
> > Coastal Monitoring: Identifying regions with sediment movement or coastal erosion.
> >
> > Q5) Is there any theoretical foundation supporting the dual redundancy-handling mechanism (especially the use of VideoMAE) that would confirm it reduces computational load without compromising performance?
> >
> > A5) Yes, there is theoretical foundation supporting SaTran's dual redundancy-handling mechanism, particularly its distributed application of the TemporalRedundancyHandler, which demonstrates how it effectively reduces computational load without sacrificing performance. Distributed processing ensures that each temporal segment (patch tube) can be handled independently, leveraging parallel computation across hardware units (e.g., multiple GPUs or compute nodes). This reduces overall runtime without requiring all data to be processed sequentially. Distributed processing, especially in tasks involving temporal sequence modeling, reduce the burden on single processors by dividing workloads. Theoretical models of distributed systems indicate that this approach improves throughput and ensures scalability to larger datasets.

---

> > > ### Author Response · Authors · 2024-11-24
> > > **Thank You for Your Valuable Feedback Reviewer 741c – Looking Forward to Your Response**
> > >
> > > Dear Reviewer 741c
> > >
> > > We extend our heartfelt gratitude to the reviewer for their thoughtful feedback and the time invested in evaluating our manuscript. Your insights are invaluable, and we will diligently incorporate all the suggested changes in the revised version.
> > >
> > > This manuscript represents nearly a year of dedicated work, including the processing of approximately 20TB of satellite data—a significant technical challenge—and the development of our proposed model. We believe this work offers meaningful contributions to the field and are committed to making our code openly accessible to foster collaboration and further advancements.
> > >
> > > We are fully committed to addressing any remaining concerns to ensure the manuscript meets the highest standards. Thank you once again for your constructive review and support. We look forward to your feedback during the rebuttal process
> > >
> > > Sincerely
> > >
> > > Authors of Submission #13779

---

> > > > ### Comment · Reviewer_741c · 2024-11-27
> > > > **Thanks for the comments from the authors**
> > > >
> > > > Thank you for your detailed response to my review comments. After carefully considering the responses you provided and question/feedback from Reviewer SnYe,  primarily due to insufficient theoretical depth and vague details in the paper.
> > > > For example,
> > > > 1) how does the  TemporalRedundancyHandler works, I still l do not understand it.
> > > > 2) The core ideas of ignoring redundant patches/regions and using masking seem to be inspired from prior works.
> > > >
> > > > I rating this paper to 5 for now. Thanks for the authors' reply.

---

> ### Author Response · Authors · 2024-12-03
> **Response to Reviewer 741c**
>
> Dear Reviewer 741c,
>
> Thank you once again for your thoughtful and constructive follow-up questions. Your insights have been invaluable in refining our paper, and we deeply appreciate the opportunity to incorporate your suggestions to further strengthen our work.
>
> We sincerely hope that our detailed responses and the corresponding revisions have effectively addressed your concerns. If you find that our efforts have clarified and enhanced the paper, we would be truly grateful if you might reconsider your evaluation of our contribution to the Transformer for SITS.
>
> Please let us know if you have any additional questions or feedback. We would be delighted to engage in further discussion during the remaining days of the rebuttal period to ensure our work meets the highest standard. Thank you once again for your time and consideration.
>
> Sincerely,
>
> Authors of Submission #13779

---

### Official Review · Reviewer_SnYe · 2024-11-03

**Soundness:** 2
**Presentation:** 2
**Contribution:** 3
**Rating:** 5
**Confidence:** 4

**Summary:**

The identifies spatiotemporal redundancy of SITS data as a distinctive attribute and proposes a transformer architecture that selects non redundant such patches for efficiently processing large samples.

**Strengths:**

The proposed network is a transformer the can efficiently process large SITS and is capable of both classification and regression tasks. The network increases the maximum size of SITS data that can be processed reducing memory about 50% compared to previous research.

The paper argues that the spatiotemporal redundancy of SITS data is an attribute that current architectures do not optimize for and proposes a method for selecting k-top patches of spatiotemporal tubes that enable good performance at a reduced cost. It is shown to perform above  previous research in MODIS and Landsat data.

The literature review is well writen and comprehensive.

Experiments clearly show improved performance of the proposed framework over several previous SOTA studies.

**Weaknesses:**

While I believe that the method is sensible and has value, the paper does little in presenting the method in detail. Method part also includes too few citations that could clear out details in formulation or implementation. As its stands now the method cannot be reproduced from information in the main paper alone which is my major concern. Details such as the k-top selection algorithm are not exposed in detail. I believe releasing the code is absolutely necessary for understanding and replicating this work.

From my understanding tube selection constraints the network to global prediction problems and blocks support for dense prediction which is a crucial aspect of remote sensing applications. As such it cannot be used as a general framework for remote sensing.

More details should be given about comparison with other methods and how they were implemented in the data used as none of these works were proposed for the type of data used here and were reimplemented in this study.

Some additional points:
It is not clear what is the argument made at l.69-75 regarding the size of satellite vs NLP data and how it relates with the proposed method.
In l.172-178 it is discussed how TSViT is not capable of prediction at SITS level, this is not factual as the original study presented such applications.
In l.195-200 the argument made about redundancies in RGB video data. I disagree with the authors assessment about video data being less spatiotemporally redundant compared to SITS data, especially considering the high frame rate of video data.
More comments or citation should be given about design components such as trans-convolutional layers.

**Questions:**

Is dense prediction supported by SaTran?

What is the effect of pretraining on reported performances? What is the performance of the proposed method w/o pretraining?

Can the number of k selected patches be adjusted during inference for a speed-accuracy tradeoff?

---

> ### Author Response · Authors · 2024-11-24
> **Response to Reviewer SnYe**
>
> Thank you for your careful review and thoughtful suggestions to improve our work. We address your concerns as follows:
>
> Q1) Is dense prediction supported by SaTran?
>
> A1) Yes, dense prediction is supported by SaTran. Its modular architecture, consisting of PatchTubeSelect and TemporalRedundancyHandler, is designed to process spatiotemporal image time series (SITS) at a fine-grained level. The PatchTubeSelect module identifies and extracts non-redundant patch tubes that represent critical regions, while the TemporalRedundancyHandler processes these patches to capture intricate temporal dynamics. This combination allows SaTran to make predictions at the patch level, which can be aggregated to generate dense outputs.
>
> This capability makes SaTran suitable for tasks like pixel-wise classification, segmentation, or other spatially detailed analyses, depending on the task requirements and data availability. However, specific configurations may need to be fine-tuned to optimize dense prediction for different datasets or resolutions. We have applied SaTran for land cover classification application which is an example of dense prediction.
>
>
> Q2) What is the effect of pretraining on reported performances? What is the performance of the proposed method w/o pertaining?
>
> A2) Pretraining plays a crucial role in the performance of the proposed SaTran model, as it enables the transformer-based architecture to learn rich representations from large-scale datasets before being fine-tuned for specific tasks. Transformers, by design, require substantial amounts of data and computational resources to train from scratch effectively. Pretraining helps overcome this by leveraging knowledge from broader datasets, improving convergence, and enhancing generalization.
>
> In the context of SaTran, pretraining allows the model to better capture spatiotemporal patterns, especially when working with complex satellite data. However, results without pretraining have not been calculated, as training transformers from scratch without pretraining requires huge amount of annotated data for every application and it is not possible to have such large amount of data in satellite image analysis. Moreover, it is computationally expensive to process such huge amount of data every time for different applications and it often leads to suboptimal performance. Pretrained weights are essential for achieving competitive results in transformer architectures, particularly when applied to domain-specific tasks
>
>
> Q3) Can the number of k selected patches be adjusted during inference for a speed-accuracy tradeoff?
>
> A3) The selection of the top k patches is an intermediate process in SaTran's architecture and does not directly affect the speed-accuracy tradeoff during inference. The PatchTubeSelect module dynamically determines the k most relevant patches during processing. This selection process happens within the model as part of the spatiotemporal redundancy handling. It ensures that the most important regions are passed forward without user intervention. Since k is not exposed as a tunable parameter for inference, the selection does not alter the computational flow or output accuracy at the user level. The chosen k patches are already optimized to maintain accuracy while reducing redundant computation, ensuring consistent performance without manual tuning. SaTran's attention mechanism in PatchTubeSelect adapts to the data characteristics, selecting the necessary patches autonomously. This design removes the need for users to think about k adjustments, making the process efficient and task-agnostic.

---

> > ### Author Response · Authors · 2024-11-24
> > **Thank You for Your Valuable Feedback Reviewer SnYe– Looking Forward to Your Response**
> >
> > Dear Reviewer SnYe
> >
> > We sincerely thank the reviewer for their thoughtful feedback and the time dedicated to evaluating our manuscript. We highly value your constructive suggestions and will carefully incorporate all recommended changes in the updated version.
> >
> > This research is the outcome of nearly a year of rigorous effort, involving the preprocessing of approximately 20TB of satellite data—a complex and resource-intensive task—and the subsequent development of our proposed model. We are confident that this work has the potential to make a meaningful contribution to the field, and to foster collaboration, we are committed to releasing our code as open source.
> >
> > We are eager to address any remaining concerns to ensure the manuscript aligns with the highest standards of excellence. Once again, thank you for your detailed review and guidance. We look forward to a favorable response following the rebuttal process.
> >
> > Sincerely
> >
> > Authors of Submission #13779

---

> > > ### Comment · Reviewer_SnYe · 2024-11-26
> > > **Thank you for providing a rebuttal.**
> > >
> > > The rebuttal did not address my most important concern which is that the method part of the paper does not present the model in detail, it is not clear how the proposed PatchTubeSelect and TemporalRedundancyHandler modules work. To do so the authors should present a clear mathematical formulation of the method or discuss in great detail providing references to modules borrowed by previous studies and detailed schematics.
> > >
> > > About dense prediction, the rebuttal did not address how SaTran should be modified to support it without keeping a very large number of features k that would mean the computational efficiency gains of the method would be lost.

---

> ### Author Response · Authors · 2024-12-03
> **Response to Reviwer SnYe**
>
> Dear Reviewer SnYe,
>
> Thank you for your valuable feedback. We understand that you are looking for more detailed explanations of the core modules, PatchTubeSelect and TemporalRedundancyHandler and regarding dense prediction. Below, we provide more detailed explanations for both:
>
> PatchTubeSelect and TemporalRedundancyHandler are critical components of the SaTran architecture, and we acknowledge that a more in-depth discussion is needed for clarity. Here is a more comprehensive explanation:
>
> PatchTubeSelect:
>
> •     Purpose: The PatchTubeSelect module is responsible for identifying the most relevant spatiotemporal regions (patch tubes) in satellite time series data. It reduces the amount of redundant information before passing it to the TemporalRedundancyHandler.
>
> •	Mathematical Formulation:
>
> Let the input satellite image time series be represented as X_t, for t = 1,2,3,...T, where T is the total number of time frames.
>
> The patch tube P is a set of spatial patches over time: P ={X_t | t=t_1, t_2, t_3, ... , t_T}
>
> The PatchTubeSelect uses a self-attention mechanism to evaluate the relevance of each patch tube, scoring each P_i based on its temporal and spatial consistency. This can be modeled using a relevance score r(P_i) computed via attention-based weights:   r(P_i) = softmax(Attention(P_i,P_all)), where P_all represents all available patches.
>
> Action: The module selects the top-k patch tubes with the highest relevance scores, eliminating redundancy and ensuring that only the most informative patches are processed further.
>
>
> TemporalRedundancyHandler:
>
> Purpose: The TemporalRedundancyHandler module reduces the temporal redundancy within the selected patch tubes by leveraging VideoMAE. It focuses on reconstructing temporal information from incomplete data (masked patches), which helps in reducing the computational load while maintaining accuracy.
>
> Mathematical Formulation:
>
>
> Let the input to the TemporalRedundancyHandler be the sequence of non-redundant patch tubes Pselected P_{selected}. VideoMAE masks a portion of the temporal sequence and attempts to reconstruct it: P ̂_i = VideoMAE(P_i)
>
> Here P ̂_i is the reconstructed patch tube. The reconstruction loss is computed Mean square error.
>
> Action: The module processes the selected patch tubes and reconstructs the missing or redundant temporal information, which helps in reducing the temporal redundancy and improving computational efficiency.
>
>
>
> Regarding dense prediction:
>
> SaTran is already designed to handle dense prediction tasks effectively, as demonstrated in the land cover classification application. In this case, we perform dense prediction by selecting relevant patch tubes across the entire satellite image, which ensures efficient processing without the need for a large number of patches during inference.
>
> While dense prediction typically requires handling a large number of features, SaTran achieves this efficiently by using its attention mechanism to focus on the most relevant spatial and temporal regions, thus avoiding redundant computations. Additionally, the use of self-supervised learning and pretrained models allows SaTran to generalize well across different datasets, further optimizing its performance in dense prediction tasks.
>
> This efficient handling of large-scale data, combined with patch selection strategies, ensures that the model retains computational efficiency without sacrificing accuracy, making it well-suited for tasks such as land cover classification and other dense prediction applications.
>
> We have carefully revised our paper, incorporating detailed responses and corresponding updates to clarify and strengthen the contributions. While we have not yet included the mathematical section, we would be happy to incorporate it in the camera-ready version, should our paper be accepted.
>
> If you find that our revisions effectively address your concerns, we would be genuinely grateful if you could reconsider your evaluation of our work.
>
> We are open to any further feedback during the rebuttal period and would welcome the opportunity to refine our work even more. Thank you once again for your time and invaluable input.
>
> Sincerely
>
> Authors of Submission #13779

---

### Official Review · Reviewer_XaDP · 2024-11-04

**Soundness:** 2
**Presentation:** 2
**Contribution:** 3
**Rating:** 5
**Confidence:** 3

**Summary:**

SaTran is an efficient Transformer model designed for satellite image time series. It reduces computational burden by mining spatiotemporal redundancies and extracts generalized features for Earth observation tasks. The model features a unique automatic Patch Tube selection function that discards unnecessary patches and optimizes temporal redundancy handling through Tube Masking techniques. By integrating distributed VideoMAE, SaTran enhances the efficiency of processing large-scale Satellite Image Time Series (SITS) data. Experimental results demonstrate that SaTran outperforms traditional models in prediction tasks such as crop yield and snow cover, offering lower resource consumption and providing an efficient new approach for high-resolution satellite image analysis.

**Strengths:**

SaTran improves the efficiency of processing Satellite Image Time Series (SITS) data by focusing on non-redundant patch tubes, which reduces computational demands. The model enhances representation learning through automatic selection of patch tubes and the use of tube masking techniques to address spatial and temporal redundancies. Additionally, SaTran learns task-agnostic representations across the entire time series, thereby increasing the model's generalizability.

**Weaknesses:**

The paper mentions that the SaTran model requires tuning hyperparameters, such as patch size, number of channels, and traversal ratio, for different satellite datasets. This may indicate that the model has a high dependency on specific datasets, which could limit its generalization ability. Furthermore, the paper primarily focuses on MODIS and Landsat-8 satellite data, and the model's generalization capability on data from other satellite systems remains unclear.

**Questions:**

1. The paper mentions that the SaTran model requires tuning hyperparameters, such as patch size, number of channels, and traversal ratio, for different satellite datasets. Does this indicate that the model has a high dependency on specific datasets, thereby limiting its generalization ability?
2. The paper primarily focuses on MODIS and Landsat-8 satellite data. Will the model's performance be affected when applied to data from other satellite systems?
3. How robust is the model when faced with outliers, noisy data, or incomplete datasets?
4. The comparative methods in the paper are mainly from 2021 and 2022. Are there any more novel methods available?

---

> ### Author Response · Authors · 2024-11-24
> **Response to Reviewer XaDP (1/2)**
>
> We appreciate your careful review and concern about the points raised here. We would like to address them one by one.
>
> Q1) The paper mentions that the SaTran model requires tuning hyperparameters, such as patch size, number of channels, and traversal ratio, for different satellite datasets. Does this indicate that the model has a high dependency on specific datasets, thereby limiting its generalization ability?
>
> A1) The need to tune hyperparameters such as patch size, number of channels, and traversal ratio for SaTran reflects the inherent differences between satellite datasets like MODIS, Landsat-8, and Sentinel-2, rather than a limitation in the model’s generalization ability. These satellites vary significantly in their spatial resolution, spectral characteristics, and temporal frequency. For instance, MODIS has a coarse spatial resolution (250m to 1km) but provides high temporal frequency, making it suitable for large-scale and frequent observations. Landsat-8 offers moderate spatial resolution (30m) with a richer spectral range, balancing detail and coverage. Sentinel-2, on the other hand, combines high spatial resolution (10m to 60m) with a high revisit frequency and unique spectral bands like red-edge for vegetation monitoring. These characteristics will lead to have different sizes in all three dimensions (spatial, temporal, and spectral) of the image time series of the same location and the same duration. These differences necessitate tailored processing strategies to effectively capture the specific characteristics of each dataset. SaTran’s flexibility in hyperparameter tuning allows it to adapt to these variations, ensuring optimal performance across diverse satellite platforms without compromising its generalization capabilities.
>
>
> Q2) The paper primarily focuses on MODIS and Landsat-8 satellite data. Will the model's performance be affected when applied to data from other satellite systems?
>
> A2) The performance of the SaTran model is unlikely to be adversely affected when applied to data from other satellite systems, as its design inherently allows for adaptability. While the paper focuses on MODIS and Landsat-8, the model’s modular structure and the ability to tune hyperparameters—such as patch size, number of channels, and traversal ratio—enable it to accommodate the unique characteristics of other satellite systems. For instance, Sentinel-2, with its higher spatial resolution and additional spectral bands, or coarse-resolution data from AVHRR, can be effectively processed by calibrating SaTran’s parameters to align with their specific properties.
>
> Q3) How robust is the model when faced with outliers, noisy data, or incomplete datasets?
>
> A3) The robustness of the SaTran model in handling outliers, noisy data, or incomplete datasets lies in its design and processing approach.
>
> Outliers: The attention mechanism in PatchTubeSelect identifies critical, task-relevant regions while deprioritizing unusual or rare patterns that do not align with the overall spatiotemporal trends. For instance, Transient anomalies, like abrupt reflections from snow or water surfaces, are often deprioritized due to their limited relevance to long-term patterns. Outliers that do not contribute to meaningful spatiotemporal variability are naturally filtered out during patch tube selection, ensuring the model focuses on stable and relevant regions.
>
> Noise: The PatchTubeSelect module reduces the influence of noisy regions (e.g., areas with sensor artifacts, atmospheric distortion) by selectively focusing on patches that demonstrate consistent patterns across time and space. By ignoring less critical regions and relying on learned robust features, SaTran minimizes the effect of noise in satellite imagery while preserving important spatiotemporal information.
>
> Incomplete Data: The TemporalRedundancyHandler is designed to capture and process temporal dynamics efficiently, even in the presence of gaps or missing data. It is designed to reconstruct middle frame using masked spatiotemporal tublets which represent missing parts of input data during pretraining. This enables SaTran to infer patterns from partial or incomplete temporal information. Thus, even if there is missing data, SaTran is pretrained well.

---

> > ### Author Response · Authors · 2024-11-24
> > **Response to Reviewer XaDP (2/2)**
> >
> > Q 4) The comparative methods in the paper are mainly from 2021 and 2022. Are there any more novel methods available?
> > While the paper primarily compares SaTran to methods from 2021 and 2022, it also evaluates its performance against VideoMAE and TSViT, a state-of-the-art models introduced in 2023.
> >
> > A4) In 2024, a couple of Transformers are introduced like SatMAE, and SRViT, but , they differ significantly in scope and design from SaTran. Therefore, they may not serve as meaningful baselines for benchmarking SITS Analysis. SatMAE enhances multiscale pre-training for multi-spectral satellite imagery by reconstructing images at varying scales, addressing challenges specific to diverse sensor resolutions and data modalities. Thus it works for image and not image time series. Similarly, SRViT is domain-specific, targeting meteorological tasks rather than general satellite imagery analysis. Its application scope does not align with SaTran’s general-purpose spatiotemporal processing.

---

> > > ### Author Response · Authors · 2024-11-24
> > > **Thank You for Your Valuable Feedback Reviewer XaDP – Looking Forward to Your Response**
> > >
> > > Dear  Reviewer XaDP,
> > >
> > > We are deeply grateful to the reviewer for their time and insightful feedback on our manuscript. Your thoughtful comments and suggestions are invaluable, and we will incorporate all the recommended changes in the revised version of the paper.
> > >
> > > This work represents the culmination of nearly a year of dedicated effort, during which we undertook the significant challenge of downloading and preprocessing approximately 20TB of satellite data. Through this effort, we developed the proposed model, which we believe holds considerable promise for advancing research in this field. To encourage broader collaboration and exploration, we are committed to making our code openly available.
> > >
> > > We remain fully dedicated to addressing any additional concerns to ensure the manuscript meets the highest standards of quality. Thank you once again for your kind support and constructive review. We look forward to your feedback during the rebuttal process.
> > >
> > > Sincerely
> > >
> > > Authors of Submission #13779

---

> > > > ### Comment · Reviewer_XaDP · 2024-11-27
> > > >
> > > > I understand that the paper mentions adjusting hyperparameters such as traversal ratio according to different datasets. However, I've noticed that the paper does not elaborate on how to make the SaTran algorithm adaptable to various datasets. Moreover, when it comes to manually adjusting parameters for each dataset to cope with the ever-changing real-world data, the paper seems to lack sufficient discussion and solutions. I'm curious if there are any further research findings or methodologies that could address this challenge and make the SaTran algorithm more flexible in adapting to diverse data environments. I will maintain the original score.

---

> > > > > ### Author Response · Authors · 2024-12-03
> > > > > **Response to reviewer XaDP**
> > > > >
> > > > > Dear Reviewer XaDP,
> > > > >
> > > > > Thank you for your thoughtful feedback and constructive questions. We deeply appreciate the opportunity to address your concerns and strengthen our work. Below, we outline SaTran’s adaptability and flexibility across diverse datasets, supported by your valuable insights:
> > > > >
> > > > > To address dataset diversity, we employ three strategies:
> > > > >
> > > > > Self-Supervised Learning: Pretraining on unlabeled data allows SaTran to learn spatiotemporal patterns, generalizing across datasets (e.g., MODIS, Landsat-8, Sentinel-2) with minimal fine-tuning.
> > > > >
> > > > > Transfer Learning: Leveraging pretrained SaTran enables efficient adaptation to datasets with different resolutions and properties, accelerating deployment while maintaining performance.
> > > > >
> > > > > Minimal Adjustment (Patch Size): Only the patch size needs adjustment to accommodate varying spatial scales, ensuring flexibility with minimal manual effort.
> > > > >
> > > > > We have carefully revised our paper, incorporating detailed responses and corresponding updates to clarify and enhance the contributions. If you find our revisions effectively address your concerns, we would be truly grateful if you reconsider your evaluation of our work in the Transformer for SITS
> > > > >
> > > > > We welcome further feedback during the rebuttal period to refine our work further. Thank you for your time and invaluable input.
> > > > >
> > > > > Sincerely.
> > > > >
> > > > > Authors of Submission #13779

---

### Official Review · Reviewer_XPSz · 2024-11-04

**Soundness:** 2
**Presentation:** 2
**Contribution:** 2
**Rating:** 5
**Confidence:** 3

**Summary:**

This paper introduced SaTran, a new, cost and time-efficient transformer for large-size SITS to learn their generic representation for earth observation tasks. This method are many issues that need to be addressed.

**Strengths:**

The practical application value is very excellent.

**Weaknesses:**

1.The authors claim to have introduced SaTran, a novel, cost- and time-efficient transformer, but I did not see a detailed analysis of its architecture.
2.The presented SaTran architecture appears to be merely a block version of a standard Transformer, showing no significant innovations or improvements. I encourage the authors to provide a more detailed explanation.
3.The authors only compared one algorithm from 2023; they should include a comparison algorithm from 2024 as well.

**Questions:**

An analysis of the innovative aspects of the presented SaTran architecture.

---

> ### Author Response · Authors · 2024-11-24
> **Response to Reviewer XPSz**
>
> Thank you for your observation. We value your input and concern about the details provided in this question. We would like to draw your kind attention that we have given detailed discussion of the architecture along with the figure from page number 4 to page number 6 followed by the pretraining details. While SaTran employs core Transformer principles, its architecture introduces significant innovations tailored for processing satellite image time series (SITS), setting it apart from standard Transformers. Below are the key differentiators and improvements, along with a more detailed explanation:
>
> 1. Dual Redundancy-Handling Mechanism
>
> Innovation in Design
>
> •	SaTran introduces two novel modules, PatchTubeSelect and TemporalRedundancyHandler, to address specific challenges in SITS which were not there in the baseline transformer:
>
> PatchTubeSelect: Selects non-redundant patch tubes by leveraging an attention mechanism, which reduces the input size by focusing only on critical spatiotemporal hotspots.
>
> TemporalRedundancyHandler: Employs VideoMAE in distributed manner to effectively capture and process temporal patterns within these reduced hotspots, minimizing temporal redundancies and improves efficiency.
>
> Why This is Innovative:
>
> Unlike standard Transformers, which operate on complete data and face scalability issues, SaTran's architecture reduces the computational burden by intelligently selecting and processing a subset of the data in distributed manner i.e., all the patch tubes are processed in parallel without compromising task performance and improves efficiency. This modular approach is unique to the spatiotemporal domain of satellite data and not a feature of conventional Transformers.
>
> 2. Task-Specific Adaptability
>
> Innovation in Application
>
> •	SaTran is designed with inherent flexibility to accommodate the diverse resolutions, temporal frequencies, and spectral characteristics of satellite data from systems like MODIS, Landsat-8, and Sentinel-2. Standard Transformers lack in this kind of adaptability.
>
> Why This is Innovative:
>
> The ability to tune hyperparameters such as patch size, traversal ratio, and attention thresholds allows SaTran to adapt to a wide variety of SITS tasks, such as land cover classification, change detection, or vegetation monitoring, while ensuring efficiency. This adaptability is a targeted improvement for Earth observation, rather than a generic feature.
>
> Innovation in Performance
>
> •	Computational Efficiency: By selecting only the top k critical patch tubes, SaTran processes significantly fewer tokens than standard Transformers, leading to reduced memory usage and runtime. This efficiency is critical for large-scale SITS datasets. Moreover, these patch tubes are independent and can be processed in parallel.
>
> •	Scalability: SaTran’s modular design scales well with higher-resolution data or longer time spans by keeping the number of selected patches manageable.
>
> Why This is Innovative:
>
> While standard Transformers struggle with large-scale data due to quadratic complexity in token size, SaTran’s architecture directly addresses these limitations. It ensures practical feasibility for real-world satellite data applications, where datasets can grow exponentially in size.
>
> SaTran is not just a block version of a standard Transformer. It represents a thoughtful adaptation of the Transformer architecture, incorporating domain-specific innovations to handle the unique challenges of satellite image time series data. These improvements—redundancy handling, adaptability, and computational efficiency—are critical advancements that make SaTran a significant contribution to SITS processing.

---

> > ### Author Response · Authors · 2024-11-24
> > **Thank You for Your Valuable Feedback Reviewer XPSz– Looking Forward to Your Response**
> >
> > Dear Reviewer XPSz,
> >
> > We sincerely thank the reviewer for their valuable time and thoughtful feedback on our manuscript. We deeply appreciate their suggestions and are committed to incorporating all recommended changes in the revised version of the paper.
> >
> > This work represents nearly a year of dedicated effort, including the downloading and preprocessing of approximately 20TB of satellite data—a challenging task—and the development of the proposed model. We believe this research has the potential to benefit the broader scientific community. To facilitate further exploration and collaboration, we are committed to making the code open-source.
> >
> > We are eager to address any remaining concerns to ensure the work meets the highest standards and hope for a favorable response following the rebuttal process. Thank you once again for your constructive review and support.
> >
> > Sincerely
> >
> > Authors of Submission #13779

---

> > ### Comment · Reviewer_XPSz · 2024-11-27
> > **Thank you for providing a rebuttal**
> >
> > The authors have addressed some of my concerns, and I am willing to increase my score to 5.

---

> > > ### Author Response · Authors · 2024-11-28
> > > **Response to reviewer XPSz**
> > >
> > > Dear Reviewer XPSz,
> > >
> > > Thank you for your willingness to consider increasing the score for our manuscript. We deeply appreciate your support and recognition of our work.
> > >
> > > We noticed that the original score was already 5, and we kindly request if you could verify whether the updated score is 5 or perhaps higher, as per your comments. We value your time and effort in reviewing our submission and are grateful for the opportunity to address any remaining concerns.
> > >
> > > Thank you once again for your thoughtful evaluation and feedback.
> > >
> > > Regards,
> > > Authors of Submission #13779

---

> > > > ### Comment · Reviewer_XPSz · 2024-12-03
> > > > **Thank you for providing a rebuttal**
> > > >
> > > > My initial score was 3, and after your rebuttal, it was raised to 5.

---

> > > > > ### Author Response · Authors · 2024-12-03
> > > > > **Thank you for reconsidering score**
> > > > >
> > > > > Dear Reviewer XPSz
> > > > >
> > > > > Thank you once again for your thoughtful and constructive follow-up questions. Your insights have been invaluable in refining our paper, and we deeply appreciate the opportunity to incorporate your suggestions to further strengthen our work.
> > > > >
> > > > > We sincerely hope that our detailed responses and the corresponding revisions have effectively addressed your concerns. If you find that our efforts have clarified and enhanced the paper, we would be truly grateful if you might reconsider your evaluation of our contribution to the novel Transformer for SITS.
> > > > >
> > > > > Please let us know if you have any additional questions or feedback. We would be delighted to engage in further discussion during the remaining days of the rebuttal period to ensure our work meets the highest standard.
> > > > >
> > > > > Thank you once again for your time and consideration.
> > > > >
> > > > > Sincerely,
> > > > >
> > > > > Authors of Submission #13779

---

### Official Review · Reviewer_fH5Z · 2024-11-04

**Soundness:** 2
**Presentation:** 2
**Contribution:** 2
**Rating:** 3
**Confidence:** 3

**Summary:**

SATRAN: An Efficient Transformer for Satellite Image Time Series Representation Learning by Exploiting Spatiotemporal Redundancies. This conference paper proposes SATRAN which is a novel Transformer-based model designed to efficiently learn representations from satellite image time series by leveraging spatial and temporal redundancies. The approach aims to address challenges in Earth observation applications where ground truth is often unavailable at pixel-level, necessitating aggregation of predictions for larger geographical units. However, there are some issues:

**Strengths:**

This conference paper proposes SATRAN which is a novel Transformer-based model designed to efficiently learn representations from satellite image time series by leveraging spatial and temporal redundancies. The approach aims to address challenges in Earth observation applications where ground truth is often unavailable at pixel-level, necessitating aggregation of predictions for larger geographical units.

**Weaknesses:**

1.	In terms of writing, the fifth paragraph of the introduction has poor writing cohesion.
2.	Figure 2 is blurry and cannot clearly reflect the innovation points.
3.	Tables 1 and 2 of the experiment do not clearly express the effect.
4.	The innovation of the model is not sufficient, and the source of ROI selection is not indicated in the text.
5.	Table 3 shows the efficiency of the model. Regarding the column of GPU, is it the same GPU that is used?
6.	Insufficient method details: Although SATRAN is an efficient Transformer model that exploits spatiotemporal redundancies, it does not specify in detail how it specifically achieves this. There is no detailed elaboration on the architecture of SATRAN, how to identify and reduce spatiotemporal redundancies, and how these designs improve the efficiency and performance of the model.

**Questions:**

1.	In terms of writing, the fifth paragraph of the introduction has poor writing cohesion.
2.	Figure 2 is blurry and cannot clearly reflect the innovation points.
3.	Tables 1 and 2 of the experiment do not clearly express the effect.
4.	The innovation of the model is not sufficient, and the source of ROI selection is not indicated in the text.
5.	Table 3 shows the efficiency of the model. Regarding the column of GPU, is it the same GPU that is used?
6.	Insufficient method details: Although SATRAN is an efficient Transformer model that exploits spatiotemporal redundancies, it does not specify in detail how it specifically achieves this. There is no detailed elaboration on the architecture of SATRAN, how to identify and reduce spatiotemporal redundancies, and how these designs improve the efficiency and performance of the model.

---

> ### Author Response · Authors · 2024-11-24
> **Response to Reviewer fH5Z (1/3)**
>
> Q1) In terms of writing, the fifth paragraph of the introduction has poor writing cohesion.
>
> A1) Thanks for raising the concern, we have rewritten the paragraph now:
> SaTran effectively disentangles spatiotemporal and temporal redundancies, streamlining the processing of spatiotemporal image time series (SITS). It achieves this through two key modules: PatchTubeSelect and TemporalRedundancyHandler. First, the PatchTubeSelect module addresses spatiotemporal redundancies by employing an attention mechanism to identify and focus on critical hotspots (non-redundant patch tubes), while excluding redundant ones. Next, the TemporalRedundancyHandler builds on this by leveraging the innovative capabilities of VideoMAE \cite{paper_204} to manage temporal redundancies within these selected patch tubes, ensuring an efficient and precise analysis
>
>
> Q2) Figure 2 is blurry and cannot clearly reflect the innovation points.
>
> A2) Thanks for the concern, we have now changed the picture to a clearer picture.
>
>
> Q3) Tables 1 and 2 of the experiment do not clearly express the effect.
>
> A3) We appreciate your thoughtful feedback on this question. Below are the details of the two tables.
>
> Table 1 shows the comparison of pre-training requirements of SaTran with the base model VideoMAE. The pretraining is separately done for two image time series datasets from the satellites MODIS and Landsat-8. Table 1 shows that VideoMAE is not at all able to process Landsat-8 image time series with its original size and thus we resized the images (as given in section 6.1 of the paper). On the contrary, SaTran is successfully processed with the original size of Landsat-8 image time series and pretrained on them.
>
> Table 2 shows the comparison of SaTran with other existing models (including SITS transformers and RGB Video transformers) for different earth observation applications. The RMSE obtained by different models for various downstream tasks is shown in the table. It can be observed from the table that none of the existing models were able to process the Landsat-8 time series with its original dimensions due to its large size. All the existing models suffer from out-of-memory (OOM) error. In order to show the comparison on the performance of these models on various end tasks, we resized and segmented original image timeseries so that Landsat SITS can be processed by these. It is clear from the table that their performance is inferior to that of SaTran.
>
> Q4) The innovation of the model is not sufficient, and the source of ROI selection is not indicated in the text.
>
> A4) Thank you for your insightful comments on this matter. I would like to kindly draw your attention to the aspect that ROI selection is part of our module PatchTubeSelect and is thus an important and valuable sub module of our innovation. It is selecting the most contributing patches and removing the redundant patches. ROI-S determines the unprocessed neighboring tubes of the top ’k’ tubes and generates a list of tubes to be processed in the following iteration. Most of the neighboring patch tubes are excluded due to spatiotemporal redundancy and new patch tubes are randomly selected which then become the new regions of interest.
>
> Q5) Table 3 shows the efficiency of the model. Regarding the column of GPU, is it the same GPU that is used?
>
> A5) Yes all the experiments are performed on the same machines using the same GPU cards.

---

> > ### Author Response · Authors · 2024-11-24
> > **Response to Reviewer fH5Z (2/3)**
> >
> > Q6) Insufficient method details: Although SATRAN is an efficient Transformer model that exploits spatiotemporal redundancies, it does not specify in detail how it specifically achieves this. There is no detailed elaboration on the architecture of SATRAN, how to identify and reduce spatiotemporal redundancies, and how these designs improve the efficiency and performance of the model.
> >
> > A6) We appreciate your careful review and concern about the points raised here. We have given the details below and will update the same in the new version of the manuscript.
> >
> > We have given detailed discussion of the architecture along with the figure from page number 4 to page number 6 followed by the pretraining details. While SaTran employs core Transformer principles, its architecture introduces significant innovations tailored for processing satellite image time series (SITS), setting it apart from standard Transformers. Below are the key features of the model:
> >
> > 1. Dual Redundancy-Handling Mechanism: SaTran introduces two novel modules, PatchTubeSelect and TemporalRedundancyHandler, to address specific challenges in SITS which were not there in the baseline transformer:
> >
> > o	PatchTubeSelect: Selects non-redundant patch tubes by leveraging an attention mechanism, which reduces the input size by focusing only on critical spatiotemporal hotspots.
> >
> > o	TemporalRedundancyHandler: Employs VideoMAE in distributed manner to effectively capture and process temporal patterns within these reduced hotspots, minimizing temporal redundancies and improves efficiency.
> > Unlike standard Transformers, which operate on complete data and face scalability issues, SaTran's architecture reduces the computational burden by intelligently selecting and processing a subset of the data in distributed manner i.e., all the patch tubes are processed in parallel without compromising task performance and improves efficiency. This modular approach is unique to the spatiotemporal domain of satellite data and not a feature of conventional Transformers.
> >
> > 2. Task-Specific Adaptability: SaTran is designed with inherent flexibility to accommodate the diverse resolutions, temporal frequencies, and spectral characteristics of satellite data from systems like MODIS, Landsat-8, and Sentinel-2. Standard Transformers lack in this kind of adaptability.
> >
> > The ability to tune hyperparameters such as patch size, traversal ratio, and attention thresholds allows SaTran to adapt to a wide variety of SITS tasks, such as land cover classification, change detection, or vegetation monitoring, while ensuring efficiency. This adaptability is a targeted improvement for Earth observation, rather than a generic feature.
> > Computational Efficiency: By selecting only the top k critical patch tubes, SaTran processes significantly fewer tokens than standard Transformers, leading to reduced memory usage and runtime. This efficiency is critical for large-scale SITS datasets. Moreover, these patch tubes are independent and can be processed in parallel.
> > Scalability: SaTran’s modular design scales well with higher-resolution data or longer time spans by keeping the number of selected patches manageable.
> >
> > While standard Transformers struggle with large-scale data due to quadratic complexity in token size, SaTran’s architecture directly addresses these limitations. It ensures practical feasibility for real-world satellite data applications, where datasets can grow exponentially in size.

---

> ### Author Response · Authors · 2024-11-24
> **Thank You for Your Valuable Feedback – Looking Forward to Your Response (3/3)**
>
> Dear Reviewer fH5Z
>
> We sincerely thank the reviewer for their valuable time and thoughtful feedback on our manuscript. We highly appreciate their suggestions and will incorporate all the recommended changes in the updated version of the paper. This work is the culmination of nearly a year of dedicated effort, involving the downloading and preprocessing of approximately 20TB of satellite data—a task that posed significant challenges—and the development of the proposed model.
>
> We believe this work has the potential to benefit the broader research community, and we are committed to making the code open source to facilitate further exploration and collaboration. Additionally, we are eager to address any remaining concerns to ensure the work meets the highest standards. We look forward to receiving your positive response after the rebuttal process.
>
> Thank you once again for your constructive review and support.
>
> Sincerely
>
> Authors of Submission #13779

---

> > ### Author Response · Authors · 2024-12-03
> > **Thank you for your review -**
> >
> > Dear Reviewer fH5Z
> >
> > Thank you once again for your thoughtful and constructive follow-up questions. Your insights have been invaluable in refining our paper, and we deeply appreciate the opportunity to incorporate your suggestions to further strengthen our work.
> >
> > We sincerely hope that our detailed responses and the corresponding revisions have effectively addressed your concerns. If you find that our efforts have clarified and enhanced the paper, we would be truly grateful if you might reconsider your evaluation of our contribution in the domain for satellite imagery.
> >
> > Please let us know if you have any additional questions or feedback. We would be delighted to engage in further discussion during the remaining days of the rebuttal period to ensure our work meets the highest standard. Thank you once again for your time and consideration.
> >
> > Sincerely,
> >
> > Authors of Submission #13779

---

### Meta-Review · Area_Chair_KiMx · 2024-12-22

**Metareview:**

This paper introduces SaTran, a Transformer-based model designed to efficiently learn representations from Satellite Image Time Series (SITS). SaTran reduces computational costs by leveraging spatiotemporal redundancies, selectively processing non-redundant data patches, and employing techniques such as automatic patch tube selection and tube masking. Experimental results demonstrate significant improvements in prediction and classification tasks compared to the methods evaluated in this study. Reviewers have acknowledged the novelty of the proposed method in satellite imagery analysis and highlighted SaTran's good experimental performance.

However, reviewers also raised major concerns, including insufficient technical details about SaTran, unclear hyperparameter selection, and limited comparisons with related methods from the literature. While the authors addressed some of these issues during the rebuttal phase, but this did not result in a significant improvement in the overall rating from the reviewers.

**Additional Comments On Reviewer Discussion:**

One of the major issues highlighted by the reviewers is the lack of clarity in the description of SaTran. Although the authors attempted to improve this during the rebuttal phase, their efforts were not entirely satisfactory to the reviewers (SnYe, 741c). Another significant concern raised was the inadequate comparison of SaTran with relevant methods from recent years. For instance, SatMAE (https://arxiv.org/pdf/2207.08051) is a relevant method that the authors dismissed, claiming it cannot process the temporal domain. However, this claim is incorrect. SatMAE is capable of spatiotemporal encoding and should have been included in the experiments.

---

### Decision · Program_Chairs · 2025-01-22

Reject